# CGMega: explainable graph neural network framework with attention mechanisms for cancer gene module dissection

Hao Li [1,5], Zebei Han[2,5], Yu Sun[1,5], Fu Wang[2], Pengzhen Hu[3], Yuang Gao[4], Xuemei Bai[1], Shiyu Peng[1], Chao Ren[1], Xiang Xu[1], Zeyu Liu[1], Hebing Chen [1] ✉, Yang Yang [2] ✉ & Xiaochen Bo [1] ✉

Cancer is rarely the straightforward consequence of an abnormality in a single gene, but rather reflects a complex interplay of many genes, represented as gene modules. Here, we leverage the recent advances of model-agnostic interpretation approach and develop CGMega, an explainable and graph attention-based deep learning framework to perform cancer gene module dissection. CGMega outperforms current approaches in cancer gene prediction, and it provides a promising approach to integrate multi-omics information. We apply CGMega to breast cancer cell line and acute myeloid leukemia (AML) patients, and we uncover the high-order gene module formed by ErbB family and tumor factors *NRG1*, *PPM1A* and *DLG2*. We identify 396 candidate AML genes, and observe the enrichment of either known AML genes or candidate AML genes in a single gene module. We also identify patient-specific AML genes and associated gene modules. Together, these results indicate that CGMega can be used to dissect cancer gene modules, and provide high-order mechanistic insights into cancer development and heterogeneity.

The complex functions of a living cell are conducted through the concerted activity of many genes and gene products. Much of the activity of a cell is organized into gene modules: sets of genes that are coregulated to respond to different conditions[1]. Gene modules have been widely studied in cell identity dissection[2], transcription factor (TF)-gene regulation[3,4], functional genome annotation[5], disease progression[6], disease origin[7], drug repurposing[8], and cancer research[9,10]. Disease-associated genes are not randomly scattered across biological networks. Instead, they tend to be located in disease gene modules[11,12]. Dissecting the gene modules that drive disease progression enables screening for the molecules that correct the network rather than targeting peripheral downstream effectors that may not be disease modifying[13–15]. Active driver modules can trigger the hallmarks of cancer and confer fitness advantages to cancer cells[16,17].

The elucidation of cancer gene modules can substantially further our understanding of cancer development and inform the design of optimal treatments[18,19].

Ever since the development of high-throughput sequencing technologies, gene module detection methods have been a cornerstone for the biological interpretation of large gene compendia. Numerous approaches and algorithms have been proposed for the detection of gene modules through measuring gene expression[3,20,21] and across omics information[22–25]. However, the methods currently in use suffer from two main drawbacks. First, recent technologies for chromosome conformation capturing have uncovered the three-dimensional (3D) genome architecture and demonstrated its critical role in establishing gene–gene relationships[26,27]. Apart from the genome, epigenome, transcriptome, and proteome information, the

[1]Academy of Military Medical Sciences, Beijing, China. [2]Department of Computer Science and Engineering, Shanghai Jiao Tong University, Key Laboratory of Shanghai Education Commission for Intelligent Interaction and Cognitive Engineering, Shanghai, China. [3]School of Life Sciences, Northwestern Polytechnical University, Xi'an, China. [4]Department of Hematology, PLA General Hospital, the Fifth Medical Center, Beijing, China. [5]These authors contributed equally: Hao Li, Zebei Han, Yu Sun. ✉e-mail: chb-1012@163.com; yangyang@cs.sjtu.edu.cn; boxiaoc@163.com

3D chromatin structure information is indispensable for detecting gene modules, especially in cancer study. But few related works have yet investigated gene module using Hi-C data. 3D chromatin data such as Hi-C data can either be represented as gene attributes that depict the spatial features of genes, or structural relationships between genes. Thus, how to encode Hi-C data and integrate with other multi-omics data challenge the precise detection of gene modules. Second, gene modules exhibit characteristics of high-order network, and the high-order interactions regulate complex functions in biological systems[28]. For example, essential genes are densely connected hubs in gene modules[29,30]. Deciphering the high-order relationships that are embedded in gene modules remains challenging. Multi-omics features have different impacts on each gene from the same gene module. For example, many genes that play important roles in tumorigenesis are not only altered on the level of their DNA sequences, but are dysregulated through epigenetic effects or other cellular mechanisms[16,31,32]. Most current co-expression clustering or correlation-based methods are not able to assign important omics features to module genes.

Recent studies have shown the utility of deep-learning algorithms to data-driven sciences, in particular, to biological data analysis[33]. These approaches provide the ability to measure large and complex multi-omics data, and support the discovery of unanticipated relationships, and from which novel hypotheses and models can be derived and used to make predictions[34]. It is worth mentioning that graph neural network (GNN) constitutes a powerful approach for measuring graph-structured data such as biological network[35], and succeed in modeling PPIs[36] and Hi-C data[37], as well as in discovering gene modules across cellular networks[38]. GNN is capable of handling different Hi-C representations, that is, either gene attributes as node features or relationships between genes as graph edges. Moreover, we have demonstrated that GNN together with its interpretation techniques are powerful tools for dissecting high-order relationships among genome interactions[39]. Together, advances in graph deep learning make it possible to dissect cancer gene modules from a multi-omics perspective, further leveraging deep understanding of the underlying data.

Here, we present a new framework (CGMega) for dissecting cancer gene module with explainable graph attention. First, we constructed a multi-omics representation graph in which nodes were genes and edges were defined as protein–protein interactions (PPIs) between genes. Node features are the concatenation of condensed Hi-C features, promoter densities of ATAC, CTCF, the histone modifications H3K4me3 and H3K27ac, and frequencies of single nucleotide variants (SNVs), copy number variants (CNVs). Then, CGMega utilized a transformer-based graph attention neural network over the multi-omics representation graph and predicted cancer genes in a semi-supervised manner. The good performance of CGMega (area under the precision recall curve, AUPRC 0.9140) ensured downstream cancer gene module detection. Finally, we adopted the model-agnostic approach GNNExplainer[40] to interpret the contribution factors to cancer genes under the context of multi-omics, and further detected the cancer gene modules with representative features. We applied CGMega to breast cancer cell line and acute myeloid leukemia (AML) patients, and uncovered the high-order relationships between genes in cancer gene modules. Together, CGMega harnesses the recent advent of GAT over multi-omics data, and gains fundamental discovery and understanding of the hierarchy in cancer gene modules.

## Results
### Overview of CGMega framework
We proposed a new framework, CGMega, for studying cancer gene modules based on graph attention and graph interpretation technologies (Fig. 1a). CGMega leverages a combination of multi-omics data across genomics, epigenomics, protein–protein interactions (PPIs),

and especially 3D genome architecture. In CGMega, we first removed the potential effects of structural variation on Hi-C contact map and normalized it with iterative correction and eigenvector decomposition (ICE)[41], and calculated the spatial distances between genes (see "Methods"). Then, singular value decomposition (SVD) was applied on the normalized matrix to get condensed Hi-C features (see "Methods"). Simultaneously, we calculated the SNV and CNV frequencies for each gene and calculated epigenetic densities within each gene promoter (see "Methods"). Then, we constructed a multi-omics information combination graph, in which the nodes represent genes and the edges are obtained from PPIs. The features of nodes are the concatenation of condensed Hi-C features, SNV and CNV frequencies, and epigenetic densities. Notably, based on the detailed evaluation in the following section, we deployed Hi-C data as node features instead of edge features. Further, we constructed a transformer-based GAT model to predict cancer genes in a semi-supervised manner (see "Methods"). Finally, CGMega implemented the model-agnostic approach GNNExplainer to detect cancer gene modules. GNNExplainer utilizes a masking approach to detect the compact subgraph structure and a small subset of node features that have a crucial role in GNN prediction[40]. Applying GNNExplainer, we identified the subset of genes that are most influential for the prediction of cancer genes, together forming the cancer gene modules (Fig. 1b). These genes are one-hop or two-hop neighbors to cancer genes, and GNNExplainer also provides important features for each gene. To examine the robustness of interpretation results in CGMega, we repeated GNNExplainer and obtained high consistent cancer gene modules (Supplementary Fig. S1a, S1b). In sum, the output of CGMega is the probability of each gene being a cancer gene and their influential genes interpreted from GATs. Gene-specific features are also assigned to these genes and together formed the gene modules.

### CGMega is effective in cancer gene prediction
CGMega identified gene modules based on the accurate prediction of cancer genes, and we thus tested the performance of CGMega in cancer gene prediction on the MCF7 cell line (see "Methods"), a human breast cancer cell line with high-confidence multi-omics data. CGMega achieved 0.9140 AUPRC (Fig. 2a, Source Data file) and 0.9630 area under the receiver operating characteristic curve (AUROC) (Supplementary Fig. S2a). To demonstrate the advances of CGMega in cancer genes prediction task, we compared CGMega with various methods (see "Methods"), encompassing both universal models GCN, GAT, MLP, SVM, and as well as specific models designed for cancer gene classification, including MTGCN[42], EMOGI[25], and MODIG[43]. Most of the models were evaluated using the same input features, while SVM and MLP had additional PPI features generated by node2vec (N2V). By computing AUPRC, AUROC, accuracy (ACC), and F1 score, CGMega outperformed all other methods across these four metrics (Fig. 2b).

Accurately predicting cancer genes often necessitates a substantial number of labeled genes, a resource that is often limited in rare cancer research scenarios. Thus, it becomes crucial to leverage the knowledge acquired from well-studied cancer genes and apply it to the context of rare cancers, thereby enhancing their prediction. To this end, we adopted a two-step approach with CGMega. In the initial stage, CGMega was pretrained on the MCF7 cell line, allowing it to grasp fundamental patterns and characteristics prevalent in cancer genes. Following pretraining, we performed fine-tuning on other cancers, enabling CGMega to adapt and fine-tune its learned representations to the specific context of those rare cancers.

To assess the performance of transfer learning, we conducted tests on the non-pretrained CGMega (trained from scratch) and the pretrained CGMega using all labeled genes (597 positives and 1839 negatives) on the K562 cell line. The pretrained CGMega demonstrated superior accuracy and F1 score, while also exhibiting comparable AUPRC and AUROC values (Supplementary Fig. S2b). Subsequently, we

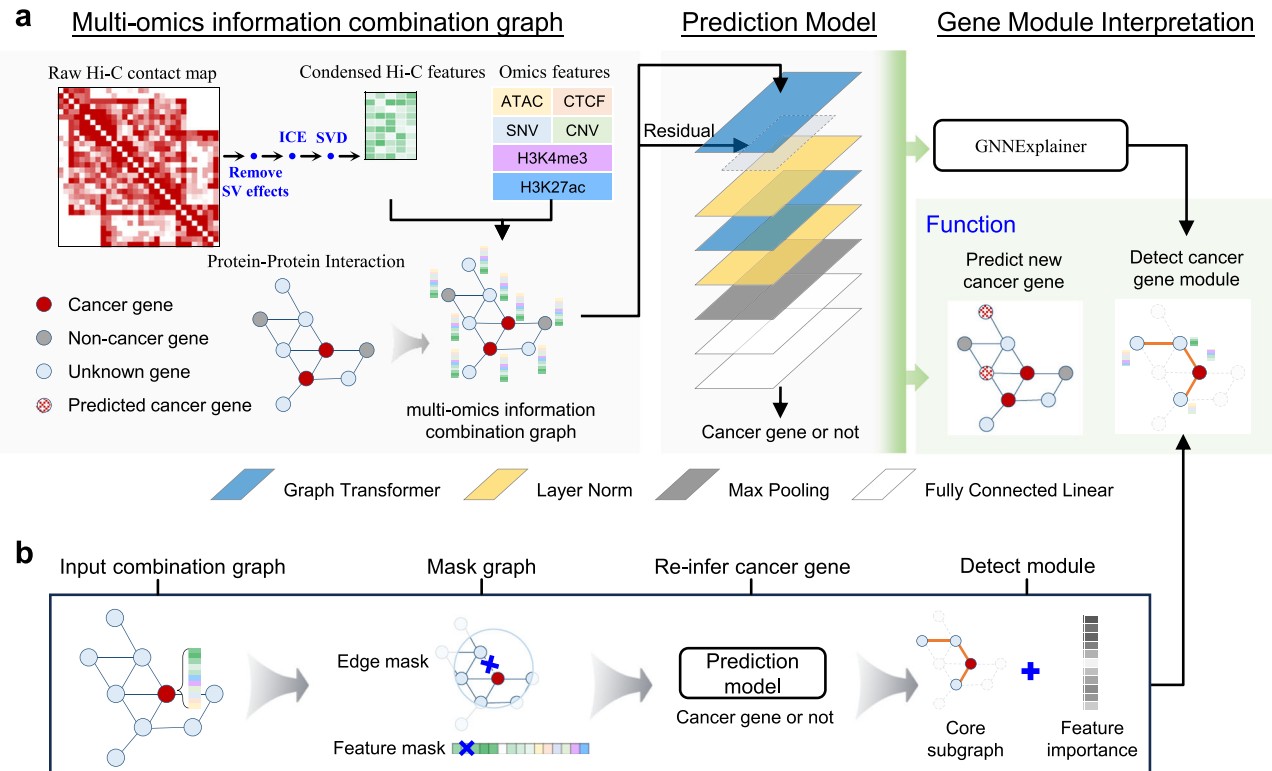

**Fig. 1 | Overview of CGMega framework. a** CGMega pipeline. First, condensed Hi-C features were obtained by removing SV effects, ICE normalization, and SVD on raw Hi-C contact map step-by-step. Simultaneously, we calculated omics features, including SNVs and CNVs frequencies for each gene as well as epigenetic densities within each gene promoter. To combine multi-omics information, we created a graph, where nodes represent genes and edges are derived from CPDB PPIs. Node features were the concatenations of condensed Hi-C features and omics features. Next, a cancer gene prediction model consisting of two Graph Transformer layers, two Layer-Norm layers, one max-pooling layer, and two fully connected linear layers was constructed. Finally, the model-agnostic approach GNNExplainer was employed to detect cancer gene modules. **b** GNNExplainer interpretation. Given a gene (represented as a node in a graph), GNNExplainer identified a subgraph G that contains the relevant features crucial for the prediction. G is a connected subgraph where the gene nodes cover at most a two-hop region with no more than 20 edges.

evaluated the non-pretrained CGMega and pretrained CGMega using downsampled labeled genes. Here, we also tested CGMega models without Hi-C features. As the number of labeled genes decreased, the performance of non-pretrained CGMega dropped sharply while the pretrained CGMega continued to have high performance (Fig. 2c, Source Data file). Moreover, the Hi-C features exhibited powerful improvements in prediction especially when the labeled genes were less than 200. Further, we compared the performance of few-shot transfer learning in CGMega with other methods, and pretrained CGMega had the highest value (Supplementary Fig. S2c).

CGMega leverages 15-dimensional gene features including 10-dimensional omics features and 5-dimensional condensed Hi-C features derived from dimensionality reduction of the Hi-C data. We performed ablation experiments by removing or shuffling gene features (Supplementary Fig. S2d), and we observed that both omics and Hi-C features made contributions for model prediction (Fig. 2d). Moreover, CGMega with 5-dimensional condensed Hi-C-only features was not as good as CGMega with 10-dimensional omics features, suggesting that the structural features may have a compensatory effect on the quality of omics features.

We tested CGMega on Hi-C data with different resolutions and read depths, CGMega maintained its stable performance using Hi-C data with resolutions from 5-kb to 25-kb, and the AUPRC slightly dropped while the Hi-C read depth decreased (Fig. 2e, Source Data file), demonstrating that our approach is robust in its adaptation to scenarios with lower data quality and holds promise for a wide range of application settings. We also tested CGMega on datasets with different ratios of positive to negative. CGMega can still achieve stable well performance with extreme ratios (Supplementary Fig. S2e) and is

better compared to other methods (Supplementary Fig. S2f). In addition, CGMega is effective for majority of the well-known PPI datasets (Supplementary Fig. S2g) and achieved better than most other methods (Supplementary Fig. S2h). We observed that the relatively poor performance of CGMega on the Multinet PPI dataset was due to its severe sparsity, and the AUPRC increased from 0.8062 (Multinet) to 0.8991 (condensed Multinet, See "Methods"). Furthermore, CGMega was also evaluated on an external dataset built with entirely new data for MCF7 cell line and achieved stable performance (Supplementary Tables 1 and 2).

## CGMega provides a new strategy for multi-omics data integration

The outperformance of CGMega benefits from the effective integration of multi-omics information, including genome, epigenome, PPIs, and especially the 3D genome architecture. Hi-C is currently the most widely used assay for investigating the 3D genome organization. However, measuring Hi-C data together with other omics data is often limited by its noise, sparsity, and variable resolution. To obtain the best performance on the cancer gene prediction task, we tested integration approaches with different Hi-C data embeddings (Fig. 3a).

Regarding Hi-C data as gene linkages. Molecular networks are important issues in biological studies[2,11,29]. For example, EMOGI has demonstrated the utility of PPIs in cancer gene prediction[25]. Hi-C data measure the interactions that connect different genomic loci and thus enables the construction of gene interaction networks. Using Hi-C contact maps, we constructed unweighted and weighted networks, respectively. In the unweighted network, interactions between genes were determined by the existence or nonexistence of contacts. For

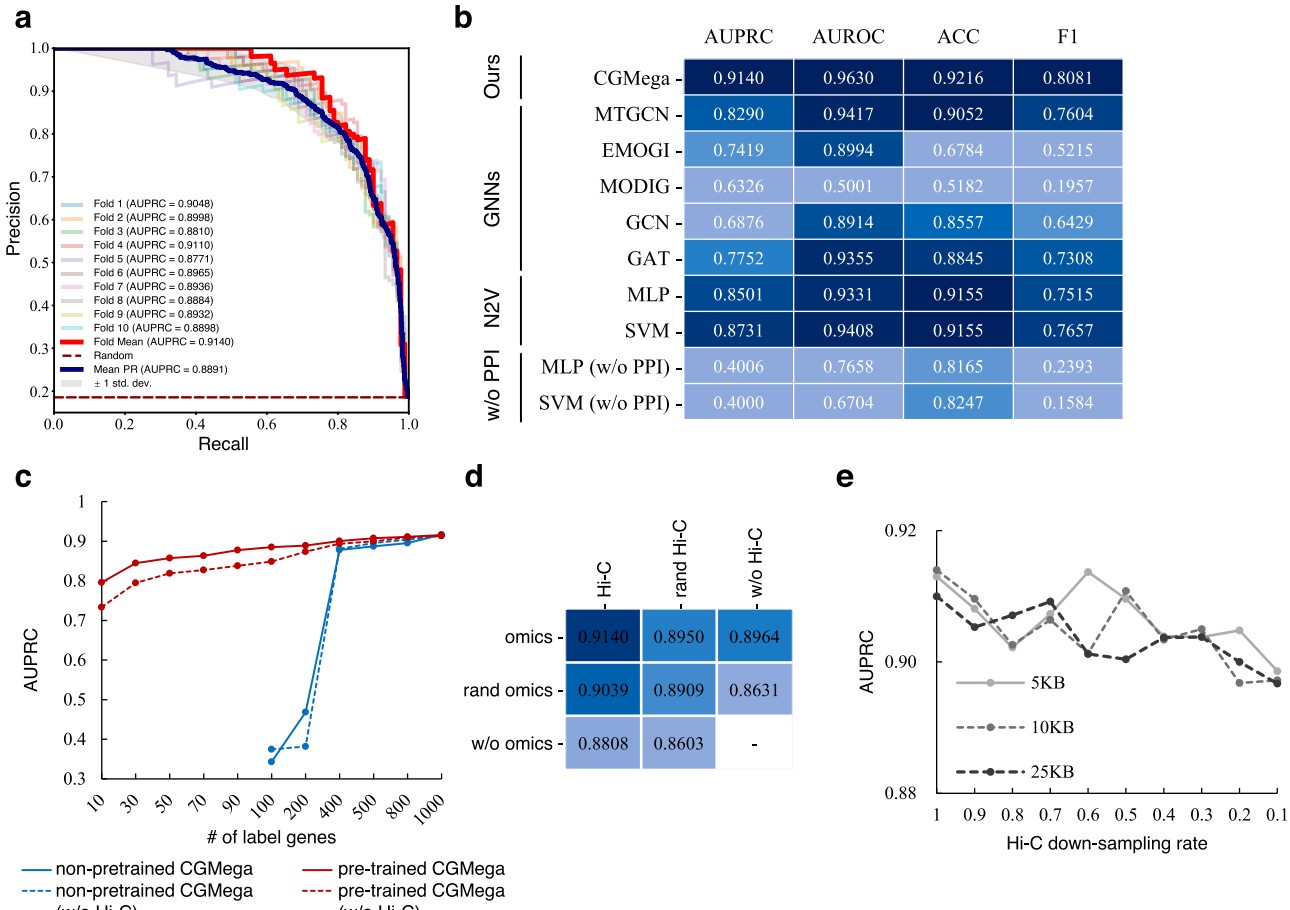

**Fig. 2 | CGMega performance in cancer gene prediction task. a** AUPRC on breast cancer cell line MCF7. **b** Methods comparison on MCF7 cell line. N2V represents node2vector. MLP and SVM were tested with and without (w/o) PPIs. **c** AUPRCs of non-pretrained CGMega and pretrained CGMega on datasets with different numbers of labeled genes. **d** Ablation experiments. AUPRCs of CGMega with random or without omics features and Hi-C features. **e** AUPRC of CGMega on Hi-C data under different resolutions and down-sampling rates. Source data are provided as a Source Data file.

weighted networks, interaction values were log10 or tenth root of contact strength. Then, epigenetic information was assigned as gene features. Finally, we combined gene interaction network with the PPI network and constructed three types of graphs: a Hi-C only graph, a Hi-C/PPI independent graph, and a Hi-C/PPI combined graph. In these, nodes represent genes and the node features are epigenetic information. We trained GAT-based neural networks on these graphs. Among these methods, the Hi-C-only graph was ineffective for predicting cancer genes (AUPRC < 0.5). The Hi-C/PPI independent graph exhibits only a marginal improvement over the PPI-only strategy. It is solely when Hi-C is combined with PPI that a modest increase, of roughly half a point, is observed in the two-edge construction methods (Fig. 3b). This result does not offer compelling support for the inclusion of Hi-C as graph structure information within the model.

Regarding Hi-C data as gene features. Hi-C data are intuitively used for measuring gene interactions. However, due to the noise and sparsity of Hi-C data, gene interaction networks based on Hi-C data tend to be incomplete and flawed. For this reason, we tested different methods of obtaining condensed Hi-C features, including Node2Vec, SVD, locally linear embedding (LLE), isometric feature mapping (ISOMAP), non-negative matrix factorization (NMF), and t-SNE. The condensed Hi-C features were concatenated with epigenetic information as gene features. PPI networks were still used to measure the interactions between genes. We also trained GAT-based neural networks on these graphs, and the situation improved significantly. Generally, incorporating Hi-C features using dimensionality reduction methods improved the prediction of cancer genes. The best-performing

method, SVD, achieved an AUPRC of 0.9140, while Node2Vec, NMF, and t-SNE also demonstrated promising results (Fig. 3c). In addition, we compared the impact of different dimensions of condensed Hi-C features for model prediction (Fig. 3d). Combining the four metrics, all methods with the Hi-C feature received a performance improvement compared to those without the Hi-C feature (Supplementary Fig. S3a). SVD-based reduction of Hi-C data to a condensed five-dimensional feature set was found to be the optimal solution based on both results.

Taken together, by systematically comparing different integration approaches with Hi-C data embedding, we showed that, in cancer gene prediction task, using Hi-C latent features as gene features outperforms measuring Hi-C data as the gene interactions directly. SVD is an effective dimensionality reduction method for combining Hi-C data with other omics data.

## Gene modules with multi-omics features in human breast cancer cell line

CGMega detects gene modules based on a model-agnostic neural network interpretation approach (Fig. 1b), and these gene modules consist of two parts: i) a core subgraph consisting of the most influential pairwise relationships for the prediction of cancer gene, and ii) 15-dimensional importance scores that quantify the contributions of each gene feature to cancer gene prediction. We applied CGMega to the human breast cancer MCF7 cell line and examined the modules of 358 known cancer genes. These cancer genes were not randomly scattered throughout gene modules; they tended to be co-located in the same modules (Supplementary Fig. S4a). This is consistent with

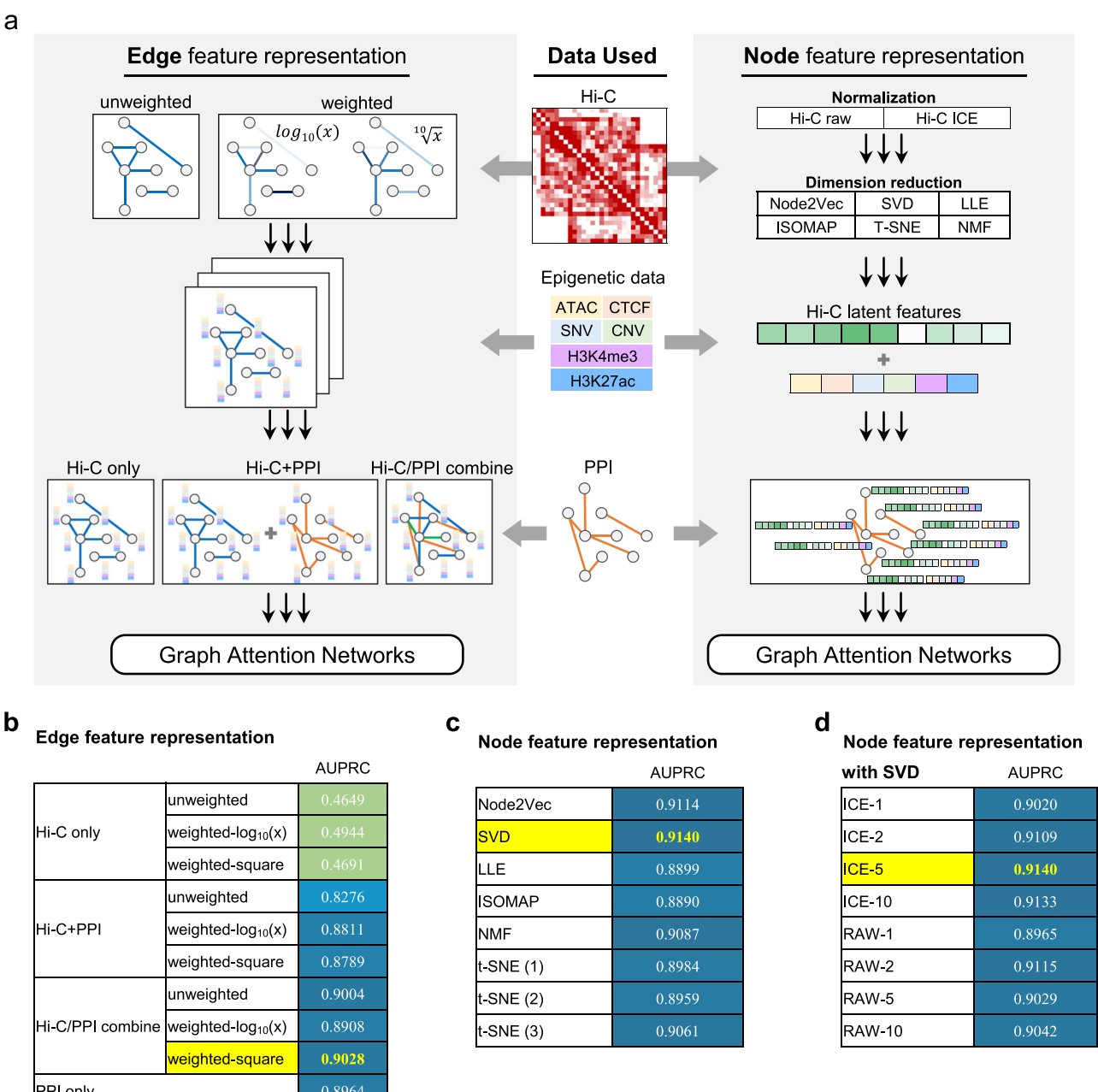

**Fig. 3 | Performance evaluation of multi-omics data integration approaches.**
**a** Design of multi-omics data integration. Left: Hi-C data were regarded as graph edges with two types. In the unweighted graph, edges were determined by the existence of Hi-C contacts or not. In the weighted graph, edge weights were interaction values calculated as log10 or tenth root of Hi-C contact maps. In either graph, node features were omics features. To combine Hi-C with PPIs, we performed GAT-based networks on three graph inputs, including (1) GAT on Hi-C graph alone, (2) two GATs on Hi-C and PPI, respectively, and then combined embeddings, and (3) first combined Hi-C graph and PPIs then performed GAT.

Right: Hi-C data were regarded as node features. Raw and normalized Hi-C data as well as different dimensionality reduction methods were tested. Then, condensed Hi-C data were concatenated with omics features, and complete node features were formed. Graph edges were determined by PPIs. For either graph structure (left or right), a GAT-based cancer gene prediction model was made, as described in Fig. 1. **b** AUPRC of cancer gene prediction model with Hi-C input as edge features. **c** AUPRC of cancer gene prediction model with Hi-C input as node features. **d** AUPRC of cancer gene prediction model with raw and normalized Hi-C.

previous reported as so-called disease modules[8]. Among these gene modules, *TP53* showed the highest enrichment and participated in 139 cancer gene modules, followed by *ESR1* (63 participations) and *AKT1* (61 participations) (Fig. 4a). In addition to these well-known cancer genes, we observed another 12 highly module-participating genes such as *XPO1*, *NCOR2*, and *PPM1A*. These genes may be the collaborators of well-known cancer genes. We also examined the structural features of gene modules with respect to their graphical metrics, including transitivity, clustering coefficients, degree centrality, and betweenness

centrality, and we found that the topological structure of cancer gene modules were significantly more consistent than that of non-cancer gene modules ($P < 2.47e-5$, paired $t$ test) (Supplementary Fig. S4b).

Beyond the topology of gene modules, we next investigated the feature importance scores. CGMega utilized 15-dimensional multi-omics features as inputs and generated an importance score for each feature. It is necessary to examine whether the importance scores were just related to the corresponding input. We thus examined the distributions of these two values, and found that feature importance is

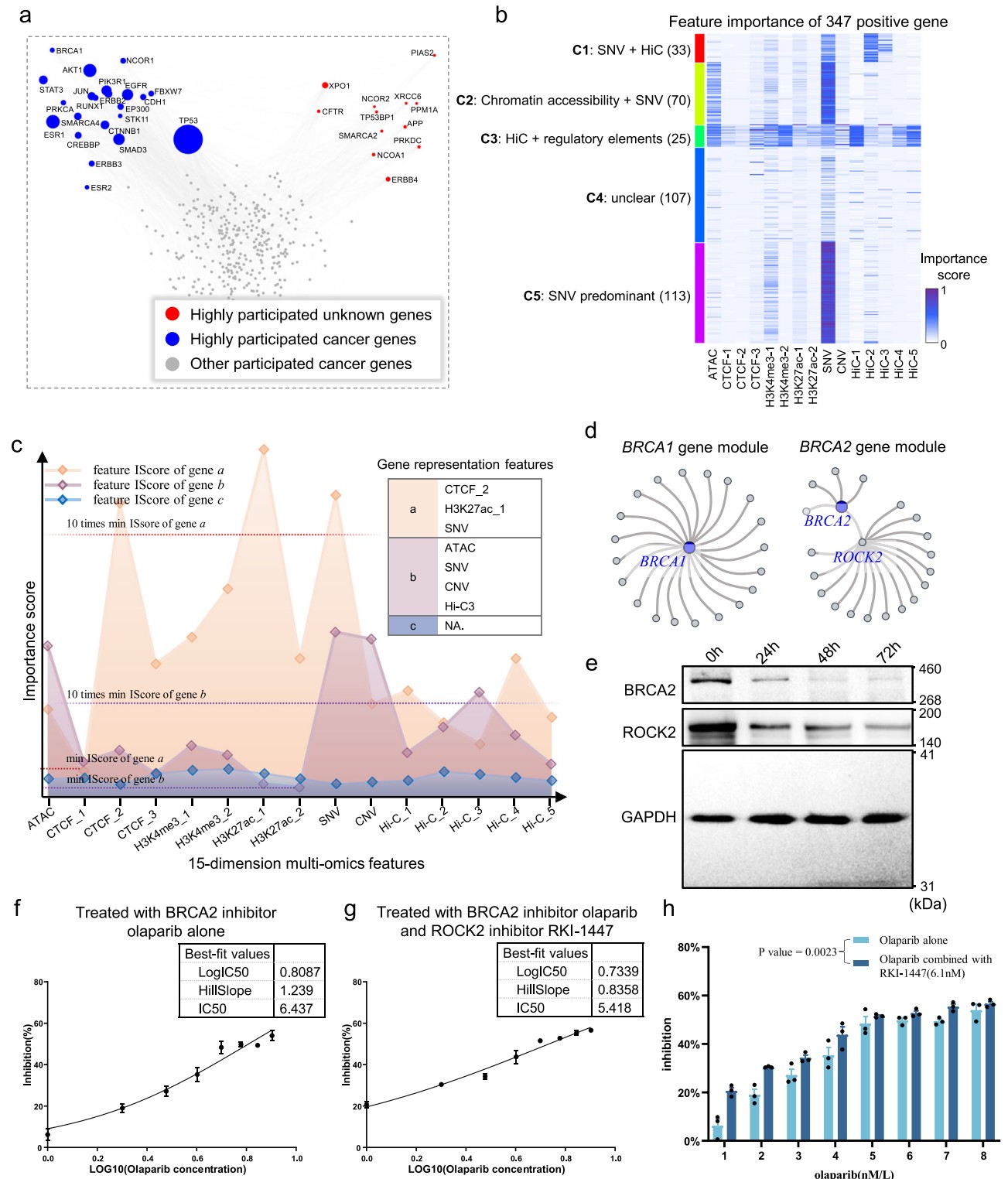

**Fig. 4 | Gene modules in breast cancer cell line. a** Scatter plot shows the gene participation in cancer gene modules. In gene modules of 358 well-known breast cancer genes, 22 known cancer genes (blue dots) and another 12 genes (red dots) which were not known as breast cancer genes were highly involved (participated in over 20 cancer gene modules). Gray dots donated known cancer genes which were not highly involved in cancer gene modules. **b** In total, 347 positive cancer genes of breast cancer were generally divided into five clusters (by K-means clustering) based on feature importance scores. **c** An example for illustrating gene representation features (RFs). For a given gene, if a feature is assigned with an importance score (calculated by GNNExplainer) 10 times higher than the minimum score,

it will be referred to as the RF of this gene. **d** Illustrations of *BRCA1* and *BRCA2* gene modules. **e** Western Blot analysis after 24 h, 48 h, and 72 h treatment. Each experiment was repeated three times independently. **f, g** Half maximal inhibitory concentration ($IC_{50}$) value of olaparib treatment (**f**) and olaparib/RKI-1447 combination treatment (**g**) after 24 hr. **h** The inhibition rate of olaparib combined with RKI-1447 is significantly higher than that of olaparib alone after 24 h treatment. Paired *t* test of two-sided was used to analyze the mean inhibition rates from two groups and the *P* value = 0.0023. **f–h** Data are presented as mean values +/− SEM and *n* = 3 biologically independent experiments were carried out for each to derive statistics. Source data are provided as a Source Data file.

irrelevant to input data (Pearson correlation coefficients $r < 0.26$, Supplementary Fig. S4c), suggesting that the importance score is the interpretation of neural networks instead of simple determination due to its input. Besides, the feature importance scores were not evenly distributed; instead, one or several features were dominant (Supplementary Fig. S4d). The feature importance scores measure the joint effect of multiple factors and help guide cancer gene classification (Fig. 4b, Source Data file). Many cancer-driven genes (class-5) were as reported to be dominated by genetic mutations. For genes in other classes, the five Hi-C features, condensed by SVD, have provided extended supplements based on their participation in each cluster: 1st, 4th and 5th Hi-C features showed joint effect with other regulatory factors on cancer driver genes (cluster-3), while 2nd and 3rd Hi-C features showed joint effect with genetic mutations (cluster-1). Some previous studies have verified these observations: (i) Gene *MYB* (in cluster-1) was reported to form fusion genes with *NFIB* due to the recurrent chromosomal translocation, which serves as a clear example of genotypic–phenotypic correlation for triple-negative breast cancer[44]. (ii) Dysregulation of gene *ADIPOR1* (in cluster-3) is widely observed in many cancers, but its genomic alteration frequencies are low[45]. This is consistent with CGMega that attributes HiC-1, HiC-5, chromatin accessibility and active histone modification H3K4me3 to *ADIPOR1*. (iii) Similar to *ADIPOR1*, gene *ALOX12* (in cluster-3) were significantly upregulated in multiple breast cancer cell lines, which protect breast cancer cell from chemotherapy-induced growth arrest and apoptosis[46,47], suggesting the importance of transcription regulation to *ALOX12*. (iv) Despite these isolated evidences, we collected RNA-seq data of breast cancers from TCGA project and identified differentially expressed genes (DEGs). The proportion of DEGs is the highest in cluster-3 (Supplementary Fig. S4e). Based on CGMega prediction, Hi-C together with other active regulatory elements have joint effect on these genes.

Based on the feature importance score, we proposed the representative features (RFs) as features that have top-ranked importance scores (Fig. 4c, see "Methods" for details). For example, the RF of the gene *TP53* is SNV while gene *PIK3R1*'s RF is Hi-C. Generally, 1158 genes had only one RF and 149 genes had multiple RFs (Supplementary Fig. S4f). We next focus on the gene modules of *BRCA1* and *BRCA2*, which are the most commonly encountered genes in breast cancer. As previously reported, these two cancer genes play different roles in the common pathway of genome protection[48]. We also observed topological differences between their gene modules. In brief, *BRCA1*, which is a pleiotropic DNA damage response (DDR) protein working in several stages of DDR, was also found to be widely connected with another 20 genes (Fig. 4d). By contrast, *BRCA2*, as a mediator of the core mechanism of homologous recombination (HR), was connected with other genes via *ROCK2*, an important gene that directly mediates HR repair[49]. Based on gene expression data from TCGA project, we found that *ROCK2* expression was positively correlated with *BRCA2* expression in breast tumor donors while there is no such correlation in normal breast tissue (Supplementary Fig. S4g). The co-expression of *BRCA2* and *ROCK2* in breast cancer suggest the joint effect in tumorigenesis, which may guide the effect enhancement of BRCA2 inhibitors on tumor cells. To test this hypothesis, we treated MCF7 cells with BRCA2 inhibitor olaparib[50,51] and with both BRCA2 inhibitor olaparib and *ROCK2* inhibitor RKI-1447[52]. Western Blot results have demonstrated the protein inhibition after 24-, 48-, and 72-h treatment (Fig. 4e, Source Data file). Then, we determined the half maximal inhibitory concentration ($IC_{50}$) of olaparib (Fig. 4f and Supplementary Fig. S5a, Source Data file) and olaparib/RKI-1447 combination (Fig. 4g and Supplementary Fig. S5b, Source Data file). $IC_{50}$ value of inhibitors combination was lower than that of BRCA2 inhibitor alone. Moreover, the inhibition rates of olaparib combined with RKI-1447 were significantly higher than those of olaparib alone after 24-h treatment (Fig. 4h, $P$ value = 0.0023, paired $t$ test, Source Data file). But it was

comparable between two groups after 48 h and 72 h treatment (Supplementary Fig. S5c). These results showed that the combination of BRCA2 and ROCK2 inhibitors was more effective than using BRCA2 inhibitor alone in inhibiting MCF7 tumor cells after 24-h treatment, suggesting a potential strategy for enhancing BRCA2 inhibitor sensitivity. In addition, SNV was the RF for both *BRCA1* and *BRCA2*. We also observed a high-order gene module combined from the *BRCA1* gene module and the *BRCA2* gene module through three shared genes including *TP53*, *SMAD3*, and *XPO1* (Supplementary Fig. S5d). Taken together, these indications mean that CGMega is capable of detecting the interpretable and high-order gene modules with multi-omics features.

### The complex gene module formed by ErbB family

The ErbB family, including *ERBB1* (also known as *EGFR*), *ERBB2* (also known as *HER2*), *ERBB3* (also known as *HER3*) and *ERBB4*, plays a central role in the tumorigenesis of many types of solid tumor. The members of the ErbB family are receptor tyrosine kinases (RTKs), which have an analogous structure[53]. However, their gene modules exhibit heterologous structures and none of these four ErbB genes were hubs in their gene modules (Fig. 5a). In the *ERBB1* gene module, *PTPRD* located at the center and connected *ERBB1* and other genes. The *ERBB2* and *ERBB4* gene modules shared the same center gene *DLG2*. *NRG1* located at the center of the *ERBB3* gene module and *ERBB4* was also present in this gene module. We examined the representative features of the ErbB family. Hi-C and SNV were major RFs for *ERBB2*, *ERBB3*, and *ERBB4* (Fig. 5b, Source Data file). The mechanisms of genetic alteration such as SNV in cancer development have been demonstrated previously[54–56]. The Hi-C features uncovered by CGMega suggest new insights regarding the signal and crosstalk between the ErbB family genes in the context of the chromatin structure in tumor progression.

In spite of the difference among the gene modules of the ErbB family, we observed several shared genes connecting the ErbB members and forming a complex module (Fig. 5c). *NRG1*, *PPM1A*, and *DLG2* were key connectors in this high-order module. Previous studies have demonstrated the importance of these three genes for cancer development. *NRG1* is a main physiological ligand to ErbB family and, together with *ERBB2* and *ERBB3*, can form a potent pro-oncogenic heterocomplex[57]. *DLG2* is a member of a family of membrane-associated guanylate kinase (MAGUK), and *DLG2* overexpression will affect the level of protein phosphorylation[58,59]. The protein serine/threonine phosphatase *PPM1A* is a crucial regulator of cell cycle progression in triple-negative breast cancer[60], and *PPM1A* is also an important factor in protein dephosphorylation[61–63]. By combining these isolated evidence with the high-order gene module, we proposed a hypothetical model of the gene module in maintaining protein phosphorylation homeostasis (Fig. 5d). The *NRG1* ligand binds to homo- or hetero-dimers of ErbB proteins, leading to the activation of ErbB-mediated downstream signaling pathways that mediate the activity of serine/threonine (Ser/Thr) protein kinases. Ser/Thr protein kinases and proteins encoded by *DLG2* modulate the phosphorylation of Ser/Thr proteins, while *PPM1A* mediates their dephosphorylation, together maintaining protein phosphorylation homeostasis.

### Gene module dissection in acute myeloid leukemia patients

We applied CGMega to acute myeloid leukemia (AML), a myeloid neoplasm that is characterized by differentiation blockade and clonal proliferation of abnormal myeloblasts in the bone marrow[64]. We collected multi-omics data for eight AML patients from a previous study[64]. Unlike the case of those cell lines, gene modules are heterogeneous across different patients[65], and the clinical course of AML is also highly heterogeneous[66]. Thus, we studied both patient-common and patient-specific cancer gene modules (Fig. 6a). First, we used CGMega to predict cancer genes and identified 2746 new genes in total

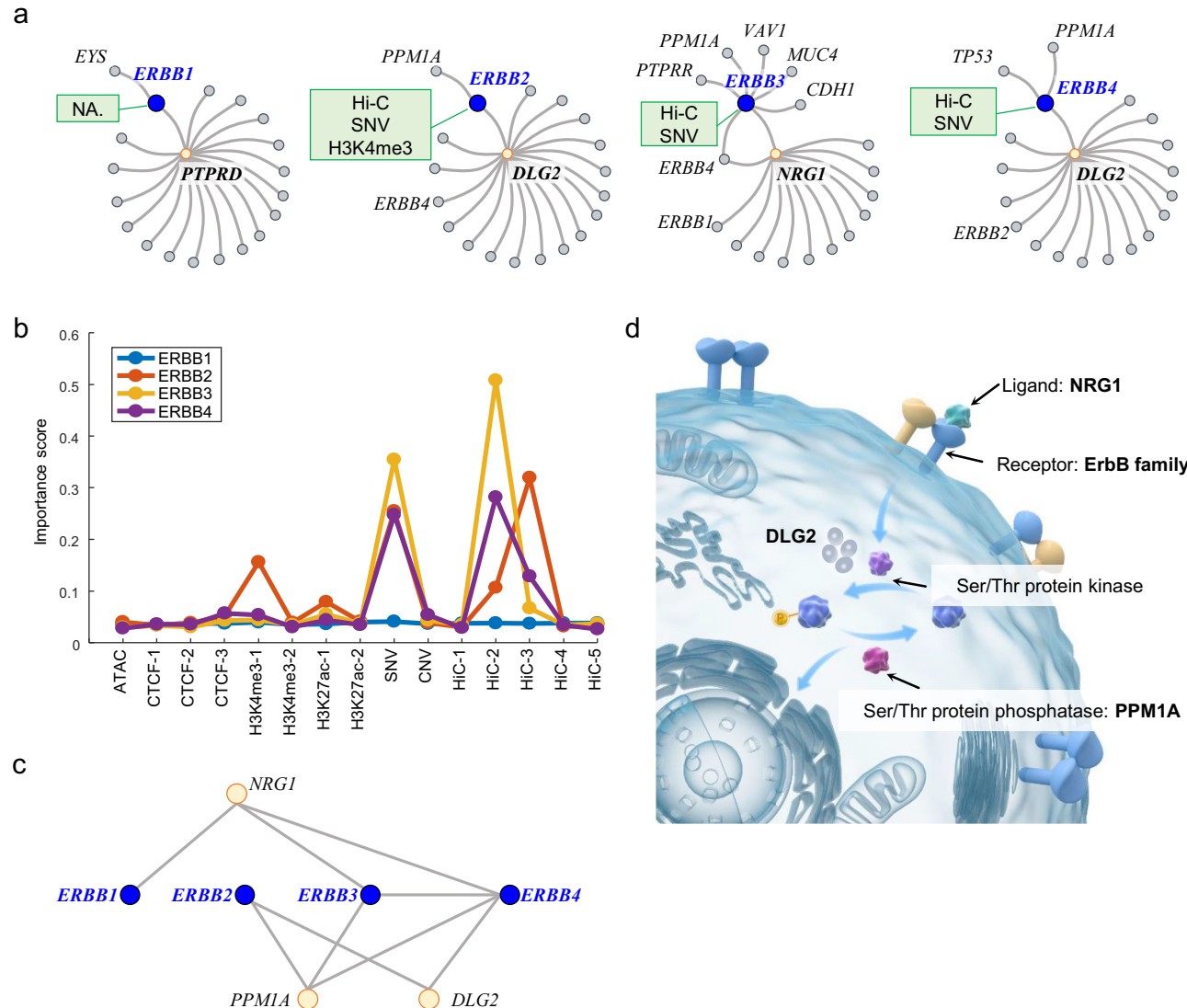

**Fig. 5 | Gene modules of ErbB family. a** Illustrations show the gene modules of the ErbB family. Blue dots indicate query genes, namely *ERBB1*, *ERBB2*, *ERBB3* and *ERBB4*. Yellow dots indicate genes located at the center of gene modules. Green boxes show the RFs of query genes. **b** Feature importance scores of the ErbB family.

**c** The high-order gene module formed by the ErbB family gene modules.
**d** Hypothetical model of the high-order gene module formed by the ErbB family gene modules in maintaining protein phosphorylation homeostasis. Details were described in the main text. Source data are provided as a Source Data file.

(Supplementary Table 3). Among these, 396 were predicted to be cancer genes in all AML patients (referred as "candidate AML genes", Supplementary Table 4). We next investigated gene functions and found that these candidate AML genes contained many essential genes and TFs, and the pan-cancer genes were significantly enriched in these 396 genes (*P* = 1.32e-22, hypergeometric test) (Fig. 6b). Moreover, Gene Ontology (GO) analysis showed that candidate AML genes together with known AML genes participated in 15 hematopoietic and blood diseases biology processes such as leukocyte migration and T-cell receptor signaling pathway (Fig. 6c). This enrichment could not be retrieved using known AML genes alone.

We then examined the AML gene modules. As with MCF7 cell line, cancer genes were also enriched in same AML gene module. This enrichment was observed not only in known AML genes but also in candidate AML genes (Supplementary Fig. S6a). In addition, 10.5% of these pairwise relationships in cancer gene modules were conserved over half of total patients. For example, in the *DLX4* gene module, connections among *DLX4*, the known cancer gene *ABL1*, and four candidate AML genes (*SP1*, *FYN*, *GRB2*, and *SMAD2*) co-occurred in multiple patients (Fig. 6d). Beyond the enrichment and co-occurrence

of AML gene modules, we observed that some candidate AML genes were shared by dozens of known AML gene modules (Supplementary Table 5). For example, *ESR1* was predicted to be candidate AML gene and it existed in modules of various known AML genes, such as *EGFR*, *PIK3CA*, and *FOS* (Supplementary Fig. S6b). This hub location implies a high-order pattern of cancer gene modules. A total of 12 known driver genes and 5 candidate AML genes were identified as hub genes, which participate in more than 20 cancer gene modules (Supplementary Fig. S6c). Among these genes, *EGFR, MYC, TP53, MAPK1*, and *PIK3R1* were well-known genes in cancer pathway[67–71]. *EP3OO, CREBBP*, and *STAT3* were used as clinical testing gene panel for myeloid tumors[72–74]. The detection of these hub genes in all AML samples demonstrates the reliability of CGMega interpretation, and suggests the potential usage as AML gene panels of those five new hub genes, including *ESR1, HDAC1, FYN, LYN*, and *GRB2*.

The AML patients used in this study come from seven different mutation types and CGMega achieved good performance (AUPRC = 0.8528 on average). We next identified patient-specific candidate AML genes for each patient (Supplementary Table 6). Examining the modules of these genes, we also observed patient-specific patterns. For

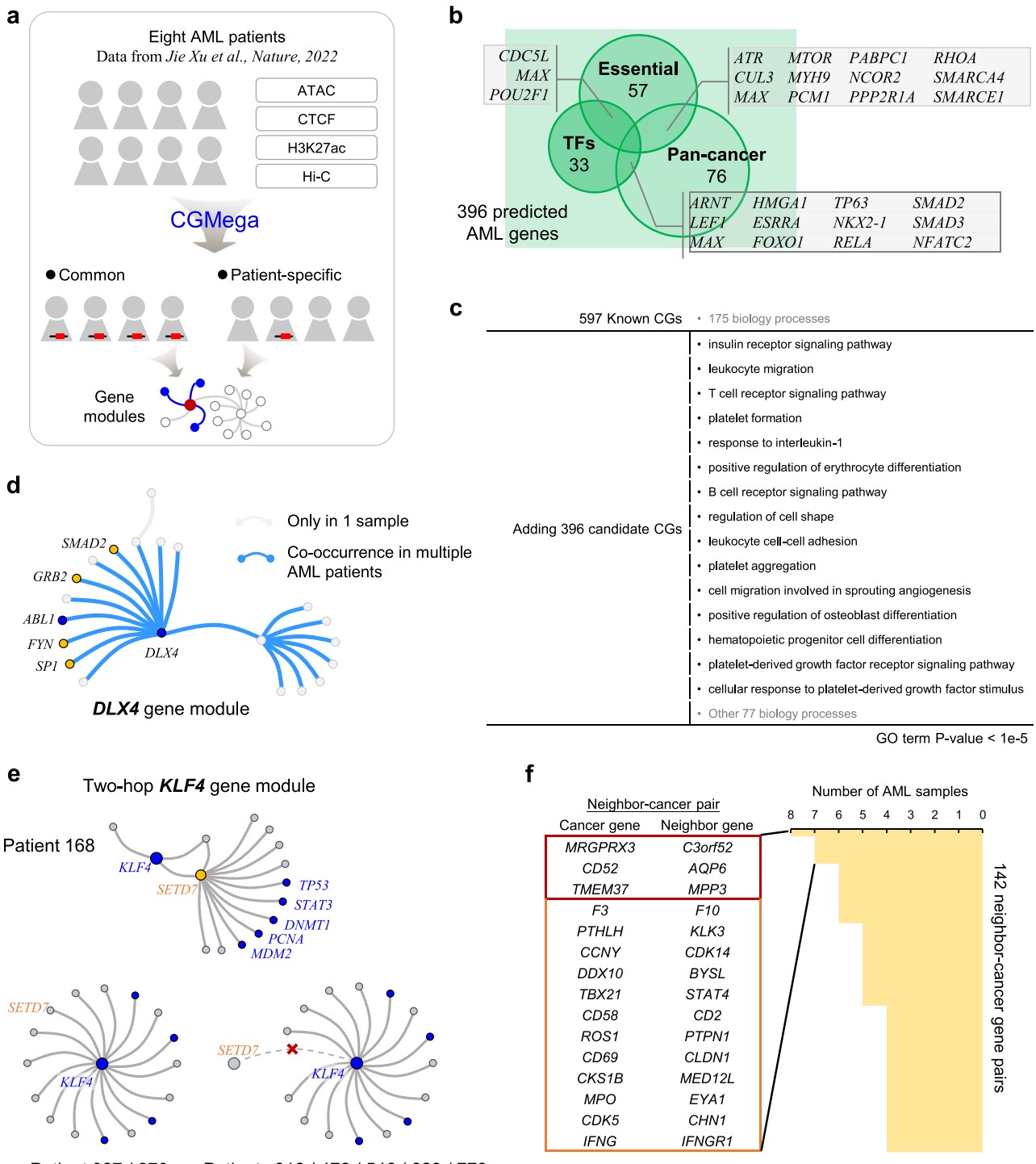

**Fig. 6 | Gene modules in AML patients. a** Application of CGMega on AML. Multi-omics data of eight AML patients were obtained from a previous study. **b** In total, 396 candidate AML genes contained essential genes, transcription factors, and pan-cancer genes. Gray boxes show genes in two categories. **c** Gene ontology (GO) enrichments. We performed GO analysis on 597 known AML genes (first line), and on 993 genes (597 known AML genes and 396 candidate AML genes), respectively. GO analysis was conducted using DAVID and GO terms with *p* value lower than 1e-5 were shown. **d** Illustration of *DLX4* gene module. Blue dots indicate known AML genes, while yellow dots indicate candidate AML genes. **e** Illustration of *KLF4* gene modules in separate patients. *SETD7* was located at the center in Patient 168 while it was just a participant in Patient 027 and Patient 270. In other patients, *SETD7* did not appear in *KLF4* gene modules. **f** We totally identified 142 neighbor-cancer gene pairs, these gene pairs were conserved in over four AML samples. Gene pairs in red box were detected in all eight AML samples and gene pairs in yellow box were detected in seven AML samples. Source data are provided as a Source Data file.

example, in the *KLF4* gene module drawn from patient 168, the candidate AML gene *SETD7* connected *KLF4* with other known AML genes including *TP53*, *STAT3*, *DNMT1*, *PCNA*, and *MDM2*. However, this two-hop gene module did not appear in other patients (Fig. 6e). Moreover, we found that the two-hop pattern was widespread in AML samples, covering about 1/3 of all AML gene modules (Supplementary Fig. S6d). The key neighbor genes, which formed neighbor-cancer gene pair in two-hop module (such as *ROCK2-BRCA2* pair in *BRCA2* gene module, Fig. 4d), provide new insights to understand tumorigenesis and drug combination strategy. We totally identified 142 such gene pairs, which were conserved in over four samples (half of the total AML samples), and found several pairs were highly conserved in all AML samples (Fig. 6f). We then performed GO analysis using both the cancer genes and key neighbor genes in these 142 pairs, and found that, different from cancer genes, the key neighbor genes were significantly enriched in signaling processes such as signal transduction and signaling pathway (Supplementary Fig. S6e), suggesting that genes participating in signal processes may be the regulator or collaborator of known cancer genes.

## Discussion

With the recent accumulation of multi-omics cancer data, different phenotypic manifestations of cancer hallmarks have seen intensive exploration[75]. However, beyond these data, the molecular mechanisms of cancer are far from transparent. The challenge of finding candidate drivers is considerable: tumors are heterogeneous, the data are noisy and highly correlated, and there are a large number of possible combinations of drivers and genes in modules[76].

Therefore, in this work, we introduced a general framework, CGMega, for predicting cancer genes and dissecting gene modules. CGMega differs from other methods in three main ways: (1) compared to current methods, CGMega is advanced in its ability to capture the 3D genome architecture, which has been widely demonstrated as a new perspective for the study of cancer[77,78]. There are three aspects of advantages by using Hi-C data: (i) Hi-C features contribute to prediction performance (Fig. 2d), AUPRC increases from 0.8964 (without Hi-C) to 0.9140 (with Hi-C). (ii) Hi-C features contribute to few-shot learning especially for training with less than 200 known cancer genes (Fig. 2c). (iii) Hi-C features embody the joint effect of multiple factors, and can be used for dissection of cancer gene modules. (2) CGMega utilizes GNNExplainer[40] to interpret contributing factors to cancer gene prediction. Compared to attention mechanism, GNNExplainer could measure various importance for the same PPI in different gene modules, and it is a model-agnostic approach for providing interpretable explanations for predictions of any GNN-based model on any graph-based machine learning task, and could avoid issues related to gradient-based methods such as LRP[79] which was used in EMOGI. Utilization of GNNExplainer help provide a comprehensive evaluation for all gene neighbors (genes that connect with predicted gene via PPIs) and molecular features (including SNVs, CNVs, histone modifications, chromatin accessibility, and 3D genome architectures) in a masking-based manner. This is critical for detecting complex and high-order gene modules. For example, *TP53* is a two-hop neighbor in the *KLF4* gene module and *SETD7* serves as the hub connecting *KLF4* and other genes (Fig. 6e). (3) CGMega shows knowledge transferability between different cancers. Previous methods, such as EMOGI, have primarily focused on pan-cancer data, neglecting the potential for knowledge transfer between different cancer types. The well performance of these methods benefits a lot from the abundant labeled genes from pan-cancer data. However, as we noted, some cancers have abundant known data, while others may not. Thus, it will be a significant achievement to explore cancer-specific driver genes. To accomplish this, we constructed pretrained model on large dataset (MCF7 cell line) and tested the performance of model fine-tuning on small dataset (sampled from K562 cell line). Furthermore, Hi-C features contribute

to this pretraining and fine-tuning strategy especially for few-shot transfer learning as trained on less than 200 known cancer genes (Fig. 2c). These results demonstrate the transferability of CGMega on different cancer types, and it is an important aspect of our study. Together, as a result, CGMega has improved performance in cancer gene prediction and cancer gene module dissection.

In addition to these advantages of CGMega, we also provided a comprehensive evaluation of approaches for Hi-C data integration with other omics data, and we demonstrated that (1) the graph structure is advanced in integrating multi-omics information, especially for molecular signals and gene relationships combination, and (2) using SVD to encode Hi-C data as gene features is better than measuring Hi-C data as gene linkages. This may be due to the high sparsity and noise of Hi-C data.

Incorporating multi-omics information enables us to consider any class of genes in a functional module and to relate the malignant phenotype to the molecular features from which it is likely to have originated. The application of CGMega on the breast cancer cell line and AML patients helped uncover that (1) cancer gene modules are widespread and well-organized including cancer gene-centered patterns (such as the *BRCA1* module in Fig. 4d) and non-cancer gene-centered patterns (such as modules of the ErbB family in Fig. 5a). (2) cancer genes (both known cancer genes and predicted ones) tend to be enriched in a single module (Supplementary Figs. S4a and S6a), suggesting the complex interplay of cancer genes in tumorigenesis. (3) beyond these well-known cancer genes, there are hub genes that are either located at the center in cancer gene modules (such as *SETD7* in the *KLF4* gene module, Fig. 6e) or are present in dozens of cancer gene modules (such as *ESR1* present in 95 cancer gene modules, Supplementary Fig. S6b). Not only can this help distinguish between driver and passenger genes, but the genes in the associated module can also provide insight into the role of the driver. (4) Combination of ROCK2 inhibitor RKI-1447 provides a potential strategy for enhancing BRCA2 inhibitor sensitivity, but more experiments are required to investigate what is the best effective time of olaparib/RKI-1447 combination and why this effect of olaparib/RKI-1447 combination disappears after 48 h. In addition, the good performance of CGMega in the breast cell line (AUPRC = 0.9140) and AML patients (average AUPRC = 0.8528) demonstrates that (1) CGMega exhibits efficacy for both cell line and donor samples, as well as for both solid tumor and liquid tumor studies. (2) CGMega is flexible for inputs with few missing molecular features (for example, a lack of SNV and CNV information in AML application), suggesting that our framework may be applicable to other similar tasks.

## Methods

### Data collection and preprocessing

Our research complies with all relevant ethical regulations. In this study, we collected well-known cancer genes and obtained mutations (SNVs), CNVs, chromatin accessibility (ATAC-seq), histone modification (H3K4me3 and H3K27ac), CTCF ChIP-seq, PPIs and 3D chromatin (Hi-C) data from public databases. The details are as follows:

**Collection of positive and negative training samples.** The collection of positive and negative samples was performed as previously reported[25]. In brief, known cancer genes were extracted from the Network of Cancer Genes (NCG)[80], Cancer Gene Census (CGC)[81] along with high-confidence (level ≥0.95) cancer genes using DigSEE[82]. Negative samples were selected using a rigorous exclusionary process, considering all protein coding genes that were not marked as positive and removing genes labeled as cancer-related or cancer candidates in NCG and KEGG (https://www.kegg.jp/kegg/kegg1.html), along with all genes from the OMIM database (https://omim.org/). In addition, genes predicted to be involved in cancer-related expressions and those with regulatory effects on oncogenes by MSigdb[83] were excluded. The remaining genes were completely unrelated to cancer (where possible

pan-oncogenes were also excluded) and were negative samples. The final dataset consisted of positive samples and negative samples obtained through these steps. Finally, 358 positive and 1581 negative samples were collected for breast cancer cell line MCF7, and 598 positive and 1838 negative samples were collected for leukemia cell line K562 and AML patients.

**Omics data.** The SNVs and CNVs information were retrieved from The Cancer Genome Atlas (TCGA) project using the corresponding data generated from the closest tumor types of breast cancer and myeloid leukemia. As for SNVs, we first filtered out the silent somatic mutations and then we calculated the mutation frequency for a gene in a sample by dividing the number of single nucleotide mutations of this gene by its genomic length. We next averaged the mutation frequency of each gene over all samples. The mean value of the obtained gene mutation frequency was magnified by 1000 times as a dimension of gene features. As for CNVs, we used the number of copy numbers of all samples of the cumulative gene and divided them by the total number of samples as the average CNV value as a dimension of gene features. For chromatin accessibility, histone modification and transcription factor binding, we downloaded ATAC-seq, H3K4me3, H3K27ac, and CTCF ChIP-seq data from the ENCODE project[84] for MCF7 and K562 cell lines, and from Gene Expression Omnibus (GEO) with accession number GSE152136 for AML patients. We then calculated the signal densities at gene promoter (TSS $\pm 1$ kb) for each marker as gene features. Human genome annotation (GRCh38) was used in this study.

**Hi-C data.** We collected Hi-C data from GEO with accession number GSE66733 for the MCF7 cell line, GSE63525 for the K562 cell line, and GSE152136 for AML patients. To remove the complex structural variants and improve accuracy, we first performed NeoLoopFinder[85] on raw Hi-C contact maps to reconstruct local Hi-C data. Then, we normalized Hi-C data using iterative correction and eigenvector decomposition (ICE)[41]. Finally, we decomposed normalized Hi-C data into five dimensions as gene features using SVD.

**Protein–protein interaction network.** We collected the PPI network from the ConsensusPathDB (CPDB)[86] database, which incorporates the interaction information from 32 widely used and important biological databases. We removed the interactions with scores (indicating experimental evidence of the interaction) lower than 0.5 and finally included approximately 270,000 interactions as graph edges. To validate the suitability of CGMega on other PPIs (Supplementary Fig. S3a), we also tested CGMega on other four PPIs including IRef, Multinet, PCNet, and STRING. These PPIs were downloaded and preprocessed as previously described[25]. Notably, Multinet dataset contains the fewest PPIs compared to other datasets, resulting in a 0.03% sparsity of CGMega input graph. We constructed condensed Multinet data by removing the 5143 isolated nodes in above Multinet input graph and generated a connected graph, in which contains 11,022 nodes and the sparsity is 0.07%.

**Dataset organization**
According to the above data collection and preprocessing, we removed genes with no corresponding information and finally obtained 16,165 protein-coding genes. Then we constructed the multi-omics information combination graph, an undirected graph, in which nodes represent genes and edges are obtained from CPDB PPI network. For the MCF7 and K562 cell lines, the node features include a total of 15 dimensions including ATAC-seq (1), CTCF (3), H3K4me3 (2), H3K27ac (2), CNV (1), SNV (1), and Hi-C (5). For AML patients, it is unavailable to get patient genomic information (SNV and CNV), and node features comprise a total of eight dimensions including ATAC (1), CTCF (1), H3K27ac (1), and Hi-C (5). For experiments on different PPI networks, graph edges were alternatively obtained from IRef, Multinet,

PCNet, and STRING. In this study, CGMega predicts cancer genes in a semi-supervised manner. Genes were labeled as positive, negative, and unknown according to the above section "Collection of positive and negative training samples". To conduct evaluation, 25% of the positive and negative genes were assigned to the test set while the remaining 75% were used for training. A tenfold cross-validation was employed on the training set to optimize hyperparameters. Using the same evaluation strategy of EMOGI, we obtained 10 models corresponding to the tenfold cross-validation, and assessed their performance on the test set, respectively. The reported metrics were derived from averaging the predictions of the 10 models on the test set. The combination graph for multi-omics information is constructed and then converted into a graphical structure dataset using PyTorch Geometric[87] for further processing and computation.

**GAT-based cancer gene prediction model**
The model consists of two graph transformer layers, two fully connected linear layers, two layer-norm layers, one max-pooling layer, and one dropout layer. The two graph transformer layers are used to perform GAT mechanism, as is discussed in the following section. The output of each graph transformer layer is passed through a corresponding layer-norm layer to normalize the activation:

$$x_i' = \frac{x_i - E(x_i)}{\sqrt{Var(x_i) + \epsilon}} \odot \gamma + \beta,$$ (1)

where $x_i$ denotes the embedding of node $i$, $\epsilon$ denotes a small constant, and $\gamma$ and $\beta$ denote learnable affine parameters.

The max-pooling layer is used to reduce the dimensionality of the features extracted by using the graph transformer layers. The two fully connected linear layers are used to make the final prediction, with the first linear layer transforming the data into a 32-dimensional representation and the second linear layer producing the final output. The dropout layer is used to prevent overfitting.

**Multi-omics feature calculation.** First, CGMega transfer original multi-omics data to input embedding according to Data Collection and Preprocessing. In detail, ATAC-seq, H3K4me3, H3K27ac, and CTCF ChIP-seq data are processed by the computeMatrix method of deepTools[88]. CNV and SNV frequency are calculated as the following equation:

$$x_i^{CNV} = \frac{\sum_{s \in S} v_{s,i}^{CNV}}{|S|},$$ (2)

$$x_i^{SNV} = \frac{\sum_{s \in S} v_{s,i}^{SNV}}{|S| * len(Gene_i)},$$ (3)

where $S$ denotes the sample set, $s$ denotes a single sample, $v_{s,i}^{CNV}$ and $v_{s,i}^{SNV}$ denote the original value of CNV and SNV to sample $s$ and gene $i$ and $len(Gene_i)$ denotes the length of gene $i$. The raw Hi-C matrix is corrected by NeoLoopFinder and ICE, and then a gene contact matrix $C$ is derived by mapping bin locations to genes:

$$C = \begin{bmatrix} c_{11} & \cdots & c_{1n} \\ \vdots & \ddots & \vdots \\ c_{n1} & \cdots & c_{nn} \end{bmatrix},$$ (4)

$$c_{ij} = \sum_{a=m'}^{m} \sum_{b=n'}^{n} h_{ab},$$ (5)

where $h_{ab}$ denotes elements in Hi-C matrix $H$, $c_{ij}$ denotes elements in gene contact matrix $C$ and gene $i$ and $j$ are located within the range $[m',m]$ and $[n',n]$ in the chromosome respectively. Finally, the gene contact matrix $C$ is decomposed to 5 dimensions using SVD. All the features after normalization are concatenated together to form the input feature matrix $\boldsymbol{X}$ as follows:

$$\boldsymbol{x}_i = [\boldsymbol{x}_i^{ATAC}, \boldsymbol{x}_i^{CTCF}, \boldsymbol{x}_i^{H3K4me3}, \boldsymbol{x}_i^{H3K27ac}, \boldsymbol{x}_i^{CNV}, \boldsymbol{x}_i^{SNV}, \boldsymbol{x}_i^{HiC}], \quad (6)$$

$$\boldsymbol{X} = [\boldsymbol{x}_1, \boldsymbol{x}_2, \ldots, \boldsymbol{x}_n]^T. \quad (7)$$

**Graph transformer layer.** CGMega employs two graph transformer layers[89] to enable graph message passing. In the graph $G = (V,E)$, the nodes are associated with feature matrix $\boldsymbol{X} \in \mathbb{R}^{n \times m}$. A node $i$ passes through a single layer as the following equation:

$$\boldsymbol{x}'_i = \text{GraphTransformerConv}(G, \boldsymbol{x}_i)$$

$$= \boldsymbol{W}_1 \boldsymbol{x}_i + \sum_{j \in N(i)} \alpha_{ij} \boldsymbol{W}_2 \boldsymbol{x}_j, \quad (8)$$

and the attention score of an edge from $j$ to $i$ can be calculated as follows:

$$\alpha_{ij} = \frac{(\boldsymbol{W}_3 \boldsymbol{x}_i)^\top (\boldsymbol{W}_4 \boldsymbol{x}_j)}{\sqrt{\boldsymbol{d}}}, \quad (9)$$

where $W_1, W_2, W_3$ and $W_4$ denote different learnable parameter matrices, $N(i)$ denotes all neighbor nodes of node $i$ and $d$ is the hidden size of each head.

To further mitigate the over-smoothing and effectively leverage the node features, CGMega also includes a residual connection for the nodes' initial features. The resulting representation learning model can be represented using the following equation:

$$\boldsymbol{x}_i^{(l+1)} = \text{GraphTransformerConv}\left(G, \left[\boldsymbol{x}_i^l, \boldsymbol{x}_i\right]\right), \quad (10)$$

where $\boldsymbol{x}_i^l, \boldsymbol{x}_i^{(l+1)}$ denotes the embedding before and after the last CGMega layer and $\boldsymbol{x}_i$ denotes the input embedding.

**Training settings.** In the experiments, a warm-up strategy for the learning rate was employed to improve stability and avoid falling into local minima during the initial training phase. The learning rate increased linearly from 0 to 0.005 for the first 20% of the total iterations. To prevent overfitting and over-smoothing, an early stop strategy was implemented that would stop the training process if the model's performance on the validation set did not improve over a consecutive 100 epochs. This strategy helps prevent the waste of computational resources and ensures that the model can achieve its optimal performance within a reasonable number of training iterations. To improve model robustness and reduce overfitting, the dropout rate was set to 0.1 for the attention mechanism and 0.4 for the other modules. The max-pooling step size was set to 2, resulting in a 32-dimensional representation. To reduce the number of parameters and ensure that training is feasible within the time and resource constraints, input graphs were sampled using neighbor sampler[90]. The subgraphs included all first and second order neighbors for each node, and training was performed on these subgraphs. Because the positive and negative samples in the dataset were highly imbalanced, especially in the breast cancer dataset, which had an approximate ratio of 1:4, 50% of the negative samples were removed from training to prevent bias towards negative samples. These hyperparameters were determined via grid search.

## Gene module interpretation

We implemented GNNExplainer to interpret the important gene features and interactions for cancer gene prediction. Given node $i$, GNNExplainer identified a subgraph $G_{Si} \subseteq G$ that contains the relevant features $X_{Si}$ that are crucial for the predicting $\hat{y}_i$. $G_{Si}$ is a connected subgraph for which the gene nodes cover at maximum a two-hop region with no more than 20 edges. The task can be formulated with the following optimization framework:

$$\max_{G_S} \mathbf{MI}(\boldsymbol{Y}, (\boldsymbol{G}_S, \boldsymbol{X}_S)) = \mathbf{H}(\boldsymbol{Y}) - \mathbf{H}(\boldsymbol{Y}|\boldsymbol{G} = \boldsymbol{G}_S, \boldsymbol{X} = \boldsymbol{X}_S). \quad (11)$$

For undirected graphs such as PPI in this work, a symmetric $M_i$ is maintained during optimization. The values for $M_i$ and $f_i$ represent the importance of corresponding edges and features, and the explanation $\boldsymbol{G}_{Si}$ and $\boldsymbol{X}_{Si}$ for the prediction $\hat{y}_i$ at node $i$ are computed as follows:

$$\boldsymbol{G}_{Si} = \boldsymbol{A}_i \odot 1\{\boldsymbol{M}_i \geq \text{edge threshold}\}, \quad (12)$$

$$\boldsymbol{X}_{Si} = \{\mathbf{x}'_j | j \in \boldsymbol{G}_{Si}\}, \quad (13)$$

where $\boldsymbol{x}'_j = \boldsymbol{x}_j \odot 1\{f_i \geq \text{feature threshold}\}$

## Methods comparison

We compared the performance of CGMega with other relevant methods. Here, we provided detailed descriptions of the model architectures and hyperparameter settings for these methods. MTGCN, EMOGI, and MODIG maintained the same model architectures as described in their original papers. For MTGCN, the dimensions of the two hidden layers were set to 300 and 100, respectively, with a linear layer dimension of 100. The dropout rate was set to 0.5. As for EMOGI, the dimensions of the two hidden layers were also 300 and 100, with a dropout rate of 0.5. The loss multiplier was set to 45.0, and the weight decay was set to 0.0005. For MODIG, the dimensions of the two hidden layers were 300 and 100 but with a dropout rate of 0.25. Omics networks from MODIG were not used in order to maintain consistency with the comparison against other methods. Both GCN and GAT adopted a two-layer architecture, with parameters kept consistent with CGMega. The MLP consisted of two linear layers, with the hidden layer and dropout settings matching those of CGMega. The optimal hyperparameters of SVM were determined through gird search. The polynomial kernel was selected, with C set to 1.0 and gamma set to the reciprocal of the product of the number of features and the variance of the input data. Though the comparison was conducted with sufficient fairness, the performance difference between MODIG and CGMega mainly derived from different datasets. MODIG was developed for pan-cancer genes while CGMega was used on cancer-specific genes, and the distinct difference between these two datasets affects MODIG performance.

## Cell viability assay

The cells were seeded in a 96-well plate, treated with an inhibitor or placebo for 24 h, 48 h, or 72 h, and cultured in an incubator with cck-8 (sigma, Germany) for 1 h. The absorbance of 450 nm was detected with an enzyme labeler (Tecan Infinite, Switzerland). Cell activity was calculated as a centenary value compared to the untreated control group. The data came from independent experiments, each of which was repeated three times. No statistical method was used to predetermine sample size. No data were excluded from the analyses. The experiments were not randomized. The investigators were not blinded to allocation during experiments and outcome assessment.

## Western blot analysis

The total proteins were obtained by lysing cells in ice-cold radio-immunoprecipitation assay (RIPA) buffer for 30 min on ice. After

centrifugation at $5180 \times g$ at 4 °C for 15 min, proteins in the super-natant were subjected to 10% or 8% sodium dodecyl sulfatepolya-crylamide gels (SDS-PAGE Gel Quick Preparation Kit, P0012AC, Beyotime, China) and then transferred to nitrocellulose membranes. The membranes containing proteins were incubated at 4 °C for 16 h with primary antibodies against BRCA2 (1:500, EPR23442-43, Abcam, Cambridge, United Kingdom, Cat No 29450-1-AP, clone name: FACD, FANCD1), ROCK2 (1:1000, sc-100425, SANTA CRUZ Biotechnology, the United States, Cat No 21645-1-AP, clone name: KIAA0619, p164 ROCK2, Rho kinase 2), GAPDH (1:30,000, 80570-1-RR, Proteintech, the United States) and then with secondary HRP-conjugated anti-mouse (1:3000, A0216, Beyotime, China), or anti-rabbit antibodies (1:3000, A0208, Beyotime, China) for 1 h at room temperature. For the validation of the above antibodies: for ROCK2, HT-1080 cells were subjected to SDS-PAGE followed by western blot with 21645-1-AP (ROCK2(middle) antibody) at dilution of 1:1000 incubated at room temperature for 1.5 h; for BRCA2, various lysates were subjected to SDS-PAGE followed by western blot with 29450-1-AP (BRCA2 antibody) at dilution of 1:4000 incubated at room temperature for 1.5 h; for GAPDH, western blot of Hela cell with anti-GAPDH (60004-1-Ig) at various dilutions. The labeled proteins were detected using immobilon Western Chemiluminescent HRP Sub-strote kit (WBKLS0500, Millipore, USA) and an LAS-4000 imaging system (Fujifilm Inc., Stanford, CT, USA). Source data are provided in the Source Data file.

### Representative features calculation

According to the feature importance scores calculated by GNNEx-plainer, we defined representative features (RFs) for each gene as features that have relatively prominent importance scores. In detail, for a given gene, among its features from ATAC, CTCF, H3K4me3 and H3K27ac, SNVs, CNVs, and Hi-C, if a feature is assigned with impor-tance score as ten times higher than the minimum score, it is referred to as the RF for this gene. We here briefly discuss how the threshold of 10-fold of the minimum was determined. At first, we explored 16 dif-ferent scenarios when the threshold was set from 5 to 20 and found that either too low or too high a threshold will lead to unreasonable results from the mechanism or scale of cancer genes. When the threshold was set to 5, the number of genes predicted as involving multiple carcinogenic mechanisms reaches over one thousand, an unreasonably high number, since it is rare that a single gene being linked to cancer through multiple ways. However, this does not mean the higher threshold will lead to more genuine calculations, because an excessively high threshold will omit the potential carcinogenic mechanisms of different genes. Therefore, the threshold of tenfold of minimum was determined as a trade-off of genuineness versus comprehensiveness.

### Identification of candidate AML genes

For each patient, unknown genes with prediction scores ranking in the top 10% were reserved as the predicted cancer genes. Then, we examined all of the predicted cancer genes across the eight patients and defined the candidate AML genes as those genes predicted as cancer genes for every patient. The corresponding modules for candidate AML genes were calculated by GNNExplainer in a manner similar to the sections of interpretation mentioned above.

### Reporting summary

Further information on research design is available in the Nature Portfolio Reporting Summary linked to this article.

### Data availability

All relevant data supporting the key findings of this study are available within the article and its Supplementary Information files. In this study, we collected data from public databases. ATAC-seq, H3K4me3,

H3K27ac, and CTCF ChIP-seq data were obtained from the ENCODE project for MCF7 and K562 cell lines, and from Gene Expression Omnibus (GEO) with accession number GSE152136 for AML patients. PPI network was obtained from the ConsensusPathDB (CPDB) data-base. Hi-C data were obtained from GEO with accession number GSE66733 for the MCF7 cell line, GSE63525 for the K562 cell line, and GSE152136 for AML patients. External datasets for CGMega perfor-mance were described in Source Data file. Source data are provided with this paper.

### Code availability

Source codes for running CGMega[91] are available on Zenodo (https://zenodo.org/records/10086978).

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

## Acknowledgements

This work was supported by the Beijing Natural Science Foundation (http://kw.beijing.gov.cn/; no. 5232025 to H.L.), Beijing Nova Program, no. 20230484290 to H.L., Independent Research Project of Medical Engineering Laboratory of Chinese PLA General Hospital, the National Natural Science Foundation of China (http://www.nsfc.gov.cn; nos. 61972251 and 62272300 to Y.Y., nos. 62173338 and 61873276 to H.C. and X.B., respectively), and the Beijing Nova Program of Science and Technology (https://mis.kw.beijing.gov.cn; no. 20220484198 to H.C.).

## Author contributions

X.B., Y.Y., H.C. and H.L. conceived the idea and H.L. designed the study. Y.S. and P.H. performed the acquisition, integration of datasets. Z.H. and F.W. developed and adapted the GAT-based cancer gene prediction and gene module detection model. Y.S. and P.H. performed model application and biological discovery. Y.S. and Z.H. implemented the Github deployment. C.R., Z.L., S.P. and X.X. performed data preprocessing. Y.G. and M.B. performed WB and tumor cell inhibition experiments. H.L. wrote the manuscript. All authors provided critical feedback and discussion, assisted in method development and application.

## Competing interests

The authors declare no competing interests.
