## [Peer Review File · Nature Communications]

CGMega: Explainable Graph Neural Network Framework with Attention Mechanisms for Cancer Gene Module DissectionReviewer #1 (Remarks to the Author):

In this research, a deep learning framework for detecting cancer gene module called CGMega is suggested. CGMega integrates multiomics information including Hi-C data to represent node features of a graph attention network, which can help improve the prediction accuracy of the model. The authors conducted several experiments to show robustness of the model, importance of omics/Hi-C features, and high performance of the model compared to the baseline methods. In addition, extensive case studies with AML and BRCA datasets show a potential of CGMega for the analysis on cancer research. The proposed framework and the manuscript can be improved in several ways as follows:

1. The analysis result (gene modules and representation features) of the proposed framework can change depending on threshold of the importance scores from GNNExplainer. As for the representation features, the threshold is 10-fold of the minimum score, which is an arbitrary one. In addition, edge and feature thresholds for gene module detection are not described in the manuscript. The authors can explain or show experimental results about an impact of thresholds on the performance of the model.
2. Proposed framework is compared with existing algorithms such as MTGCN and EMOGI in Fig. 2b. Recently, a new method called MODIG has been published and it shows better performance than MTGCN and EMOGI according to the manuscript. Please consider adding MODIG into the comparative analysis.
Zhao, W., Gu, X., Chen, S., Wu, J., & Zhou, Z. (2022). MODIG: integrating multi-omics and multi-dimensional gene network for cancer driver gene identification based on graph attention network model. *Bioinformatics*, 38(21), 4901-4907.
3. The authors showed performance of CGMega on different PPI databases in Fig. S2. It seems that performance is lower when using Iref and Multinet than other databases. It would be better if any discussion about this result (e.g. effect of the number of nodes/edges in the network) can be added.
4. In line 342-356, gene modules of ErbB family is described and three of them share the representation features as Hi-C and SNV. Although the authors added a number of references that explain the importance of SNV for cancer mechanisms (line 353-354), there should be a discussion about Hi-C representation features, as it is the main novelty of the proposed method. What would be the biological implication from the results of CGMega? Are there any common features for genes with Hi-C representation features?

Reviewer #2 (Remarks to the Author):

I co-reviewed this manuscript with one of the reviewers who provided the listed reports as part of the Nature Communications initiative to facilitate training in peer review and appropriate recognition for co-reviewers.

Reviewer #3 (Remarks to the Author):

The study's writing and organisation are comprehensive and well-executed, but the technical novelty is lacking. Specifically, CGMega's replacement of GCN with GAT, available through PyTorch Geometric, and the use of GNNExplainer for model explanation are not novel contributions with significance.

The contributions of the study are listed in the Discussion clearly. However, additional evidence is required to support the conclusions.

- The authors emphasise the usage of Hi-C data as a unique feature of CGMega. It is then not clear that if all other methods compared in Figure 2 also used Hi-C data. If yes, then using Hi-C data is not a technical novelty of CGMega. If not, then it is not clear whether the GAT-based architecture provided better prediction.
- The second contribution is the utilisation of GNNExplainer. However, EMOGI is also explainable, and it would be better to demonstrate whether GNNExplainer provide more reliable model explanation than the way EMOGI explains the model.

- The third contribution is about pretraining and fine-tuning, but I believe this is a strategy that can be applied to any neural network-based model. What is unique in the CGMega about this?

This study is comprehensive as it covered most experiments and comparisons conducted in similar studies such as EMOGI. However, it does not use any external test set for unbiased performance evaluation. Using 25% of the data as the test set is acceptable but not as strong as using an external dataset in our field. Importantly, although in the Methods it says "To conduct evaluation, 25% of the positive and negative genes were assigned to the test set while the remaining 75% were divided into 10 equal parts.", Figure 2 and S2 seem to be reporting cross-validation results rather than test set results.

The authors claim proteome data are integrated. "The outperformance of CGMega benefits from the effective integration of multi-omics information, including genome, epigenome, proteome, and especially the 3D genome architecture." I believe the authors mean the usage of PPI data for proteomics, because otherwise there is no other proteomics mentioned. However, using PPI is not generally considered as the integration of proteomics data. In the similar work EMOGI, which is also based on PPI, does not claim proteomic data are integrated.

The methodology is overall sound and can meet expected standards with some further revision and clarification, but it lacks key novelties as a method paper. For example, the formulas for GAT, graph transformer and GNNExplainer are all included in the Methods at the moment without additional linkage to omics data. In this case, the authors could simply cite the original paper as there is no need to re-write the formulas.

The data used for training/test in each analysis is not clear. For example, in the analysis related to Fig2, CGMega only uses one cell line for both training and test?

In the Data collection and preprocessing section, the author states "To validate the suitability of CGMega on other PPIs (Fig. S3a)..." (line 540). However, Fig. S3a doesn't seem to be about different other PPIs, which is a crucial comparison.

The code repository needs to be polished for the work to be reproduced.

Are you satisfied that all data and source code needed to reproduce the results of the paper have been made available?

The quality of the overall project looks good, but it still needs some revision. The tutorial notebook is great but it currently seems broken with missing datasets.

Are you satisfied that the results can be replicated using the code/software and dataset provided in the study?

I was not able to run the Tutorial notebook due to missing datasets.

Were you able to run the tool successfully?

No. First, some dependencies were missing (No module named 'torch_sparse'). I figured out by myself but then the tutorial notebook couldn't run because of missing datasets (No such file or directory: 'data/Breast_Cancer_Matrix/MCF7_Adjacent_Matrix_Ice').

Was the code sufficiently documented to allow another researcher to follow the algorithm?

There is some documentation, but they are not sufficient, especially for users who do not have experience of deep learning development. Also the current installation documentation on GitHub doesn't seem to support GPU. Additional cuda packages were required.

Can the software be run on a widely available operating system?

Although not tested, I think the software can be run on a wide range of operating systems, given the required dependencies are available across different operating systems.

To your knowledge, do available tools or software exist that perform in a similar way to the reported software?

Yes. I know the EMOGI work relatively well and I think EMOGI solves exactly the same questions as CGMega. CGMega has changed a few components in the model and shown superior predictive performance. But they are essentially very similar.

In cases when the source code is not provided but the mathematical description of the algorithm is; was the core mathematical algorithm sufficiently documented to allow another researcher to reproduce it?

Yes.

Reviewer #4 (Remarks to the Author):

The authors propose a method for predicting cancer genes by integrating multi-omics features. Besides, GNNExplainer is utilized for interpretability analysis, and some significant patterns are identified from gene modules in the context at either cell-line level or patient level. The prediction results demonstrate the effectiveness and robustness of the proposed model, and the explanations help us better understand the structure of cancer gene modules. This paper is interesting and well-written.

Major comments

1. In page 12, the authors show a case study of detecting gene modules and important features of BRCA1 and BRCA2. While the goal here is to show that the predictions are interpretable because of some important patterns, but the mechanistic explanations are not supported clearly (e.g. only an intermediate gene of BRCA2 are related to cancer mechanism). It is suggested to further show how the functions of the detected genes are related to the cancer mechanisms, by providing more in-depth explanations.
2. In page 18, the authors identified some significant high-order patterns of AML gene modules, but it's unclear how these patterns are related with the AML. Therefore, it is suggested to conduct more analyses about mechanisms of AML disease, in order to mine more insightful patterns and further prove the interpretability of the method.
3. The authors conduct interpretability analysis using GNNExplainer. Since the graph transformer layers with attention mechanism are used within the CGMega, I wonder why the authors do not use the learned attention scores of graph transformer layers for explanation. Would the analysis result be different from the explanations given by GNNExplainer?

Minor comments

1. From Figure 2e, SVM seems to perform better than MTGCN in terms of AUPRC, F1 score and ACC, so why do you say that MTGCN is the SOTA in page 6?
2. In Page 6, the sentence "we tested retrained CGMega (training from scratch) and pre-trained CGMega on K562 cell line using all labeled 164 genes" is not clear enough. I cannot figure out on which cell line the first model is retrained on and which cell line tested on, and whether the second model is fine-tuned or only is a pre-trained version. It is suggested to remove 'tested' and modify this sentence carefully.
3. In Figure S2b, it is interesting that the performance of the retrained CGMega does not exceed the pre-trained model. Can you explain more about this result?
4. In Figure S2e, you show the results of CGMega with different positive:negative ratios, but none of the results are matched with the original CGMega in Figure 2e. What is the positive:negative ratio you used for the original CGMega?
5. Will CGMega still outperform the best baseline if you use different positive:negative training ratios (in Figure S2e) or different PPI databases (Figure S2f)?

Point-by-point response to reviewers:

Reviewer #1 (Remarks to the Author):

In this research, a deep learning framework for detecting cancer gene module called CGMega is suggested. CGMega integrates multiomics information including Hi-C data to represent node features of a graph attention network, which can help improve the prediction accuracy of the model. The authors conducted several experiments to show robustness of the model, importance of omics/Hi-C features, and high performance of the model compared to the baseline methods. In addition, extensive case studies with AML and BRCA datasets show a potential of CGMega for the analysis on cancer research. The proposed framework and the manuscript can be improved in several ways as follows:

Response:

We appreciate the time and energies that the reviewer poured into our manuscript. We really thank the professional suggestions which greatly help improve the quality of our work.

According to the concerns raised by the reviewer, we improved our work from the following four aspects:

1. We have added more analysis to test the performance of our framework.
2. We have added detailed methods description and extended the discussion.
3. We have conducted inhibitor treatment experiments to investigate gene modules.
4. We have optimized and uploaded both the original data and codes onto Zenodo for further use.

All results have been added to the revised manuscript, and point-to-point responses are as below:

Q1. The analysis result (gene modules and representation features) of the proposed framework can change depending on threshold of the importance scores from GNNExplainer. As for the representation features, the threshold is 10-fold of the minimum score, which is an arbitrary one. In addition, edge and feature thresholds for gene module detection are not described in the manuscript. The authors can explain or show experimental results about an impact of thresholds on the performance of the model.

1.1 Response:

The concern raised by the reviewer about the thresholds is appreciated. We conducted threshold to restrict the size and connectivity of gene modules in order to ensure the computational feasibility for subsequent calculations. The threshold-setting does not involve strictly controlling the value of a certain importance score, but rather utilizes a top-ranking approach. Specifically, for each gene,

CGMega keeps the edges as the final explanation when the combination of the following conditions is true: (a) the linked nodes are within two-hop neighbors of the central gene node, (b) the importance scores rank in the top 20, and (c) they must form a connective graph. Thanks for this comment which reminds us that the previous manuscript failed to convey that the threshold was not related to the absolute magnitude of importance scores, and we have polished the relevant text.

As for the representative features, we added the details about how the 10-fold threshold was determined. At first, we calculated 16 different scenarios when the threshold was set from 5 to 20, respectively (Table-1 for reviewer). With a certain threshold, genes could be classified into different categories according to the type of representative features they possess, including one or multiple from ATAC-seq, CTCF ChIP-seq, Hi-C, CNV, SNV and histone modifications. For example, genes with only ATAC as the representative feature were classified as ATAC_group, genes with only Hi-C information as the representative feature were classified as Hi-C_group, and genes with multiple types of representative features were classified as Multiple_group.

Then, we compared gene sets with different representative features under different threshold values. An excessively low threshold tends to assign too many genes to the Multiple_group. When the threshold is set to 5-fold of the minimum, the number of genes in the Multiple_group reaches 1025, an unreasonably high number, since it is rare that a single gene being linked to cancer in ways of multiple mechanisms. With the increased threshold, the number of genes in the Multiple_group decreases. Whereas, this does not mean the higher threshold will lead to more genuine calculations, because an excessively high threshold will omit the potential carcinogenic mechanisms of different genes. For example, when threshold is set to 20-fold of the minimum, the number of genes that have representative features sharply drops to 245, an obviously underestimated number compared to the scale of pan-cancer genes already known (more than a few hundreds).

In summary, either too high or too low a threshold will lead to biologically unreasonable results for the scale or the mechanisms of cancer genes. Therefore, **the threshold of 10-fold of minimum was determined as a trade-off of genuineness versus comprehensiveness according to the well-established biological knowledge.**

Table-1 for reviewer

	SNV_group	CNV_group	CTCF_group	Hi-C_group	Multiple_group
Thresh_5	165	68	239	369	1025
Thresh_6	172	113	290	466	715
Thresh_7	177	134	330	517	475
Thresh_8	182	146	353	539	312
Thresh_9	190	139	352	528	208
Thresh_10	197	126	327	488	149
Thresh_11	209	112	293	417	98
Thresh_12	213	95	250	308	70

Thresh_13	211	52	201	165	52
Thresh_14	217	19	151	78	35
Thresh_15	220	8	114	36	25
Thresh_16	224	3	96	21	17
Thresh_17	219	2	81	16	13
Thresh_18	210	2	69	12	11
Thresh_19	203	2	55	10	6
Thresh_20	188	1	43	7	5

2. Proposed framework is compared with existing algorithms such as MTGCN and EMOGI in Fig. 2b. Recently, a new method called MODIG has been published and it shows better performance than MTGCN and EMOGI according to the manuscript. Please consider adding MODIG into the comparative analysis.

Zhao, W., Gu, X., Chen, S., Wu, J., & Zhou, Z. (2022). MODIG: integrating multi-omics and multi-dimensional gene network for cancer driver gene identification based on graph attention network model. *Bioinformatics*, 38(21), 4901-4907.

1.2 Response:

Thanks for reminding us to compare with MODIG. To address this concern, we detailed compared the required data and methods performance.

As shown in Table-2 for reviewer, MODIG requires more input data than CGMega. First, MODIG is trained on ~3000 pan-cancer genes while CGMega is suitable for cancer-specific genes, and CGMega supports few-shot learning with as few as 200 labelled genes (as shown in Fig. 2c). Second, CGMega required 15-dimensions node features including 10-dimensions epigenetic features and 5-dimensions Hi-C features while MODIG required 48-dimensions pan-cancer features. Third, MODIG required sufficient structure features including PPI, GO, gene co-expression, pathway, DNA sequence similarity, and semantic similarity while CGMega only import PPI as structure input.

Table-2 for reviewer

Input data		MODIG	CGMega
Gene label source		Pan-cancer genes	Cancer-specific genes
Number of labelled genes	Positive	796	358
	Negative	2187	1581
	Total	2983	1939
Node features	Somatic Mutation	√	√
	CNV	√	√
	DNA methylation	√	-
	Gene Expression	√	-
	Chromatin Accessibility	-	√

	Histone modification	-	✓
	Hi-C	-	✓
	Total Feature dimension	48	15
Structure features	PPI	✓	✓
	Gene co-expression	✓	-
	GO	✓	-
	Pathway	✓	-
	DNA sequence similarity	✓	-
	Semantic similarity	✓	-
	Total structure types	6	1

Next, we compared the performance of CGMega and MODIG on both datasets. As shown in Table-3 for reviewer, on our dataset, CGMega performed much better than MODIG. This is expected because as reported in MODIG paper, multiple structure features is very important for MODIG performance (as shown in Table-4 for reviewer, corresponding to Table 1 in MODIG paper), and CGMega dataset does not contain as sufficient structure features and multi-omics node features as MODIG required. But for MODIG dataset, MODIG achieved the best while CGMega performed better than EMOGI, MTGCN and others (See Table-5 for reviewer).

Together, when we have sufficient omics data and prior knowledge such as known biological pathway, MODIG is a good choice for pan-cancer genes prediction. CGMega is advanced in cancer-specific genes prediction especially with limited data and knowledge. In addition, the motivation of MODIG and CGMega is different. MODIG was designed for cancer driver gene prediction while we proposed CGMega to detect cancer driver gene-associated modules.

We thank the reviewer again for helping us improve our work and we have added the comparisons between CGMega and MODIG in our revised manuscript.

Table-3 for reviewer

Comparison on CGMega dataset				
	AUPRC	AUROC	ACC	F1 score
CGMega	0.9140	0.9630	0.9216	0.8081
MODIG	0.6326	0.5001	0.5182	0.1957
MTGCN	0.8290	0.9417	0.9052	0.7604
EMOGI	0.7419	0.8994	0.6784	0.5215
GCN	0.6876	0.8914	0.8557	0.6734
GAT	0.7752	0.9355	0.8845	0.7308

Table-4 for reviewer

MODIG Performance on different structure features, corresponding to Table 1 in MODIG paper					
Structure feature	Node features	AUPRC	AUROC	ACC	F1
PPI	Multi-omics	0.5955	0.8243	0.7572	0.6281

PPI+GO+Seq+Path+Exp	Multi-omics	0.8164	0.9086	0.8441	0.7293
-------------	--------	--------	--------	--------

Table-5 for reviewer

Comparison on MODIG dataset						
PPI source	MODIG	CGMega	EMOGI	MTGCN	GCN	GAT
STRING (MODIG used)	0.81	0.74	0.71	0.68	0.66	0.63
CPDG (CGMega used)	0.80	0.75	0.70	0.62	0.67	0.62

3. The authors showed performance of CGMega on different PPI databases in Fig. S2. It seems that performance is lower when using Iref and Multinet than other databases. It would be better if any discussion about this result (e.g. effect of the number of nodes/edges in the network) can be added.

1.3 Response:

Thanks for reminding us to discuss CGMega performance on different PPI datasets. Actually as the reviewer speculated, the numbers of nodes/edges have an impact on CGMega performance.

As described in “Dataset organization” section of the manuscript, the input graph for CGMega takes 16,165 protein coding genes as graph nodes, and the size of graph adjacent matrix is 16,165 x 16,165, which are extracted from PPI network. Thus, the node number and edge number of PPI datasets directly determine the graph structure for CGMega training. As shown in Table-6 for reviewer, Multinet dataset contains the fewest PPIs compared to other datasets, resulting in a sparsity of 0.03% (83766/16165²). This extreme sparsity of Multinet input graph may be the main factor for CGMega bad performance. To confirm this speculation, we removed the 5143 isolated nodes in above Multinet input graph and generated a connected graph (namely condensed Multinet). Condensed Multinet contains 11,022 nodes and the sparsity is 0.07%. When testing CGMega (See Table-7 for reviewer), the AURPC increased from 0.8062 (Multinet) to 0.8991 (condensed Multinet), suggesting the great impact of graph sparsity on CGMega performance. Notably, MTGCN and GCN outperformed CGMega on condensed Multinet dataset, suggesting that graph convolution is more suitable for small size graph modeling.

Table-6 for reviewer

Information for PPI datasets			
	Node #	Edge #	Sparsity of CGMega input graph
CPDB	12262	273765	0.10%
STRING	10967	253535	0.10%
iRef	14960	342006	0.13%
PCNet	15999	2192197	0.84%
Multinet	11022	83766	0.03%

Table-7 for reviewer

Performance on Multinet and condensed Multinet PPI datasets					
	CGMega	MTGCN	EMOGI	GAT	GCN
Multinet	0.8062	0.8545	0.6151	0.7037	0.5739
Condensed Multinet	0.8991	0.9154	0.8855	0.7912	0.9081

We collected iRef PPIs from its original version, which was published in 2008. It is the oldest dataset used in our study. As reported, protein-protein interactions in iRef were established using proteins primary sequence, taxonomy identifiers, and the Secure Hash Algorithm, which lack experimental evidence. The performance of other current methods on iRef dataset were also poor (Table-8 for reviewer).

Table-8 for reviewer

AURPC on different PPI datasets					
PPI dataset	CGMega	MTGCN	EMOGI	GAT	GCN
CPDB	0.9140	0.8290	0.7393	0.7752	0.6976
STRING	0.8953	0.8332	0.5304	0.8495	0.4955
iRef	0.8659	0.7451	0.6387	0.5919	0.5672
PCNet	0.8955	0.8462	0.8113	0.7340	0.7471
Multinet	0.8062	0.8545	0.6151	0.7037	0.5739

We have added above discussion in our revised manuscript.

4. In line 342-356, gene modules of ErbB family is described and three of them share the representation features as Hi-C and SNV. Although the authors added a number of references that explain the importance of SNV for cancer mechanisms (line 353-354), there should be a discussion about Hi-C representation features, as it is the main novelty of the proposed method. What would be the biological implication from the results of CGMega? Are there any common features for genes with Hi-C representation features?

1.4 Response:

Thanks for the helpful advice.

In this work, representation features (RFs) were calculated based on feature importance scores, which were generated to quantify the contribution of each feature. Thus, to discuss the biological implication of RFs, we first focus on feature importance scores. As shown in Fig-1 for reviewer, genes from MCF7 cell line were generally divided into five clusters (by *K*-means clustering) based on feature importance scores. **Implications among these gene clusters are:** Many cancer driven genes (class-5) were as reported to be dominated by genetic mutations. Except for these well-known cancer genes. The five Hi-C features, condensed by SVD, have provided extended supplements based on their participations in each cluster: 1st, 4th and 5th Hi-C features showed joint effect with

other regulatory factors on cancer driver genes (cluster-3), while 2nd and 3rd Hi-C features showed joint effect with genetic mutations (cluster-1). Some previous studies have verified our observations: (i) Gene *MYB* (in cluster-1) was reported to form fusion genes with *NFIB* due to the recurrent chromosomal translocation, which serves as a clear example of genotypic–phenotypic correlation for triple-negative breast cancer [1]. (ii) Dysregulation of gene *ADIPOR1* (in cluster-3) is widely observed in many cancers, but its genomic alteration frequencies are low [2]. This is consistent with CGMega that attributes HiC-1, HiC-5, chromatin accessibility and active histone modification H3K4me3 to *ADIPOR1*. (iii) Similar to *ADIPOR1*, gene *ALOX12* (in cluster-3) were significantly up-regulated in multiple breast cancer cell lines, which protect breast cancer cell from chemotherapy-induced growth arrest and apoptosis [3, 4], suggesting the importance of transcription regulation to *ALOX12*. (iv) Despite these isolated evidences, we collected RNA-seq data of breast cancers from TCGA project and identified differentially expressed genes (DEGs). The proportion of DEGs is the highest in cluster-3 (Fig-2 for reviewer). CGMega predicts HiC together with other active regulatory elements have joint effect of these genes. In addition, we also performed GO analysis on genes from cluster-2. *Positive regulation of gene expression* and *regulation of transcription from RNA polymerase II promoter* were enriched in this gene set.

Fig-1 for reviewer.

Fig-2 for reviewer.

Based on the rationality of feature importance score, we proposed RFs to focus on the most crucial information. In MCF7 cell line, 1109 genes were attributed with Hi-C as RFs. A cancer gene with Hi-C as its RF means that chromatin structure might be an indispensable driving factor in its association with cancer. **We summarized three possible ways to study cancer genes based on Hi-C RFs:** (i) Hi-C RFs reflect potential structure variations. For example, in ErbB family, three members (*ErbB2*, *ErbB3* and *ErbB4*) share Hi-C as RF while the other one *ErbB1* (*EGFR*) does not (Fig-3 for reviewer). (ii) Genes with only Hi-C as RFs account to 171. Among these genes, 144 and 22 of them locate at crucial positions as loop anchors and TAD boundaries, respectively. (iii) Genes of which the RFs include not only Hi-C, but also epigenetic and (or) structural variants information tend to link with aberrant gene expression resulted from gene fusion (*MYB*, *NOTCH1*, *CD74*, *BRAF*, *FGR3*, *NDRG1*, *RUNX1*)^[5,6,7,8], enhancer hijacking (*ABCC1*, *B3GNT9*, *FOPNL*)^[9] and so on.

Fig-3 for reviewer.

Taken together, feature importance scores provided by CGMega measure the joint effect of multiple factors, and can be used for guiding cancer genes classification. Hi-C RFs are meaningful in investigating the associations between chromatin structure and cancer gene mechanism such as dysregulation and genomic variation.

We have added above results and discussion in our revised manuscript.

References:

1. Martelotto, Luciano G et al., Genomic landscape of adenoid cystic carcinoma of the breast. *The Journal of pathology*, DOI:10.1002/path.4573
2. Zhuoyuan Chen et al., Distinct roles of ADIPOR1 and ADIPOR2: A pan-cancer analysis. *Front. Endocrinol.*, 2023, DOI: 10.3389/fendo.2023.1119534
3. Zhen Huang et al., ALOX12 inhibition sensitizes breast cancer to chemotherapy via AMPK activation and inhibition of lipid synthesis. *Biochem Biophys Res Commun.*, 2019, DOI: 10.1016/j.bbrc.2019.04.101

4. Siyuan Weng et al., ALOX12: A Novel Insight in Bevacizumab Response, Immunotherapy Effect, and Prognosis of Colorectal Cancer. *Front. Immunol.*, 2022, DOI: 10.3389/fimmu.2022.910582
5. Francisco J Novo et al., TICdb: a collection of gene-mapped translocation breakpoints in cancer. *BMC Genomics*, 2007, DOI: 10.1186/1471-2164-8-33
6. Marta Persson et al., Recurrent fusion of MYB and NFIB transcription factor genes in carcinomas of the breast and head and neck. *PNAS*, 2009, DOI: 10.1073/pnas.0909114106
7. A Fehr et al., The MYB-NFIB gene fusion-a novel genetic link between adenoid cystic carcinoma and dermal cylindroma. *the Journal of Pathology*, 2011, DOI: 10.1002/path.2909
8. Dan R Robinson et al., Functionally recurrent rearrangements of the MAST kinase and Notch gene families in breast cancer. *Nature Medicine*, 2011, DOI: 10.1038/nm.2580
9. Jie Xu et al., Subtype-specific 3D genome alteration in acute myeloid leukaemia. *Nature*, 2022, DOI: 10.1038/s41586-022-05365-x

Reviewer #2 (Remarks to the Author):

I co-reviewed this manuscript with one of the reviewers who provided the listed reports as part of the Nature Communications initiative to facilitate training in peer review and appropriate recognition for co-reviewers.

Response:

We appreciate the time and energies that the reviewer poured into testing our framework.

According to the issues from the reviewer as listed in Reviewer 3 comments, **we have optimized and uploaded both the original data and codes onto Zenodo** (<https://zenodo.org/records/10086978>). Besides, to facilitate users to quickly start using CGMega, we also provide a pre-built **docker image** (built based on Ubuntu 20.04) as well as its Readme.md on Zenodo. Dependent packages and CGMega are installed in an anaconda environment. This environment will be activated automatically when you log in.

We hope the revised version will run well and help the reviewer test CGMega's performance.

Reviewer #3 (Remarks to the Author):

The study's writing and organisation are comprehensive and well-executed, but the technical novelty is lacking. Specifically, CGMega's replacement of GCN with GAT, available through PyTorch Geometric, and the use of GNNExplainer for model explanation are not novel contributions with significance.

Response:

We appreciate the reviewer's efforts in helping us improving our work, especially for helping us highlight the novelties of our framework.

According to the issues from the three reviewers, we improved our work from the following four aspects:

1. We have added more analysis to test the performance of our framework.
2. We have added detailed methods description and extended the discussion, especially in discussing the novelties of our work.
3. We have conducted inhibitor treatment experiments to investigate gene modules.
4. We have optimized and uploaded both the original data and codes onto Zenodo for further use.

All results have been added to the revised manuscript, and point-to-point responses are as below:

The contributions of the study are listed in the Discussion clearly. However, additional evidence

is required to support the conclusions.

Q1. The authors emphasise the usage of Hi-C data as a unique feature of CGMega. It is then not clear that if all other methods compared in Figure 2 also used Hi-C data. If yes, then using Hi-C data is not a technical novelty of CGMega. If not, then it is not clear whether the GAT-based architecture provided better prediction.

2.1 Response:

Thanks for the comment. The technical novelty of CGMega lies in its ability to exploit Hi-C data for cancer gene prediction, as **there has been no prior methods for predicting cancer genes that utilizes Hi-C data to our best knowledge.**

There are three aspects of advantages by using Hi-C data: (i) Hi-C features have contributed to prediction performance (Fig-1A for reviewer, related to Fig. 2d in manuscript), AUPRC increases from 0.8964 (without Hi-C) to 0.9140 (with Hi-C). (ii) Hi-C features contribute to few-shot learning especially for training with less than 200 known cancer genes (Fig-1B for reviewer, related to Fig. 2c in manuscript). (iii) Hi-C features help measure the joint effect of multiple factors, and can be used for guiding cancer genes classification. As shown in Fig-1C for reviewer, genes from MCF7 cell line were generally divided into five clusters (by K-means clustering) based on feature importance scores. Implications among these gene clusters are: Many cancer driven genes (class-5) were as reported to be dominated by genetic mutations. Except for these well-known cancer genes. The five Hi-C features, condensed by SVD, have provided extended supplements based on their participations in each cluster: 1st, 4th and 5th Hi-C features showed joint effect with other regulatory factors on cancer driver genes (cluster-3), while 2nd and 3rd Hi-C features showed joint effect with genetic mutations (cluster-1).

In addition, we also test whether GAT-based architecture contributes to prediction. We constructed GCN-based architecture by replacing the transformer layer with GCN, the input is the same as CGMega which contains Hi-C data. As shown in Fig-2 for reviewer, **the performance of GAT-based is better than that of GCN-based architecture.**

C Feature importance of 347 positive gene

Fig-1 for reviewer

Architecture	AUPRC	AUROC	ACC	F1 score
CGMega	0.9140	0.9630	0.9216	0.8081
GCN-based	0.8927	0.9502	0.9113	0.7725

Fig-2 for reviewer

Together, both Hi-C data and GAT-based architecture contribute to CGMega performance.

Thanks for the helpful advice in emphasizing the usage of Hi-C data, and we have added the analysis and discussion in the revised manuscript.

Q2. The second contribution is the utilisation of GNNExplainer. However, EMOGI is also explainable, and it would be better to demonstrate whether GNNExplainer provide more reliable model explanation than the way EMOGI explains the model.

2.2 Response:

Thank you for your comment. EMOGI incorporates layer-wise relevance propagation (LRP), a gradient-based approach, to explain the model. **LRP is initially developed for MLP and CNN models** ^[1], and involves calculating partial derivatives of the model outputs to the input feature matrix and the corresponding adjacency matrix Laplacian matrix. As reported, gradient-based methods are often not suitable for explaining predictions made on graphs ^[2], and the efficacy of LRP hinges on the specific architecture of the model, and LRP is more difficult to apply to complex neural network architecture such as GAT. Due to these limitations, interpretations generated by LRP will be meaningless in some cases.

To avoid these limitations, we conduct GNNExplainer in CGMega, it is a model-agnostic approach for providing interpretable explanations for predictions of **any GNN-based model** on any graph-based machine learning task, and could **avoid issues related to gradient-based methods such as gradient saturation**. These issues are exacerbated on discrete inputs such as graph adjacency matrices since the gradient values can be very large but only on very small intervals. Thus, utilisation of GNNExplainer is more advanced compared to LRP in EMOGI.

We have added this statement in the revised manuscript.

References:

1. Sebastian Bach et al., On Pixel-Wise Explanations for Non-Linear Classifier Decisions by Layer-Wise Relevance Propagation. *PLoS One*, 2015, DOI: 10.1371/journal.pone.0130140
2. Ying R et al., GNNExplainer: Generating Explanations for Graph Neural Networks. *NeurIPS*, 2019, PMID: 32265580; PMCID: PMC7138248.

Q3. The third contribution is about pretraining and fine-tuning, but I believe this is a strategy that can be applied to any neural network-based model. What is unique in the CGMega about this?

2.3 Response:

Thanks for pointing out this poor discussion in our manuscript. Actually as the reviewer said, **CGMega does not employ a unique pretraining/fine-tuning scheme.** Rather, we aim to use these techniques to explore the potential for **knowledge transfer between different cancers.**

Previous methods, such as EMOGI, have **primarily focused on pan-cancer data**, neglecting the potential for knowledge transfer between different cancer types. The well performance of these methods benefits a lot from the abundant labelled genes from pan-cancer data. However, as we noted, some cancers have abundant known data, while others may not. Thus, it will be a significant achievement to explore cancer-specific driver genes. To accomplish this, we constructed pretrained model on large dataset (MCF7 cell line) and test the performance of model fine-tuning on small dataset (sampled from K562 cell line). Furthermore, Hi-C features contributes to this pretraining and fine-tuning strategy especially for few-shot transfer learning as trained on less than 200 known cancer genes (as discussed in Q1, Fig-1B for reviewer, related to Fig. 2c in manuscript). These results demonstrate the transferability of CGMega on different cancer types, and it is an important aspect of our study.

According to the helpful advice, we have revised this discussion in our revised manuscript.

Q4. This study is comprehensive as it covered most experiments and comparisons conducted in similar studies such as EMOGI. However, it does not use any external test set for unbiased performance evaluation. Using 25% of the data as the test set is acceptable but not as strong as using an external dataset in our field. Importantly, although in the Methods it says “To conduct evaluation, 25% of the positive and negative genes were assigned to the test set while the remaining 75% were divided into 10 equal parts.”, Figure 2 and S2 seem to be reporting cross-validation results rather than test set results.

2.4 Response:

Thanks for this comment. **Figure 2 and S2 do not report cross-validation results.** Here we adopted the same evaluation strategy as EMOGI, which involves dividing the training set into 10 folds and using 10-fold cross-validation to **generate 10 trained models.** We then used these 10 models to predict the test set respectively, and the averaged test results were taken as the final results. We understand that this may not have been explicitly stated in the manuscript, and we have added a description in Methods section to clarify these settings. We apologize for any confusion this may have caused.

5. The authors claim proteome data are integrated. "The outperformance of CGMega benefits

from the effective integration of multi-omics information, including genome, epigenome, proteome, and especially the 3D genome architecture." I believe the authors mean the usage of PPI data for proteomics, because otherwise there is no other proteomics mentioned. However, using PPI is not generally considered as the integration of proteomics data. In the similar work EMOGI, which is also based on PPI, does not claim proteomic data are integrated.

2.5 Response:

We apologize for this unrigorous claim. As the reviewer point out, we indeed used PPI rather than integrate proteomics data. We have corrected these claims in the revised manuscript.

6. The methodology is overall sound and can meet expected standards with some further revision and clarification, but it lacks key novelties as a method paper. For example, the formulas for GAT, graph transformer and GNNExplainer are all included in the Methods at the moment without additional linkage to omics data. In this case, the authors could simply cite the original paper as there is no need to re-write the formulas.

2.6 Response:

As the reviewer pointed out, both GAT and GNNExplainer are well defined methods, and we have simplified the original formula description and provided more task-specific formal descriptions in the revised manuscript.

Although we conduct the well-known GAT and GNNExplainer in our work, CGMega own its unique novelties. First and the most important, CGMega incorporates Hi-C data for cancer gene and gene module prediction. This enables cancer-specific gene study rather than previous pan-cancer studies, and is helpful for few-shot transfer learning (as discussed above in Q3 response). To achieve this incorporation, we test different strategies for integrating Hi-C and other omics data, and finally confirmed SVD as the best strategy. It is a novel and instructive in developing multi-omics combination methods. Second, to utilize multi-omics data and prevent overfitting, we have made modifications to the original GAT model by adding residual connections between the input feature and the last graph transformer layer (See Fig-3 for reviewer and Fig. 1a in revised manuscript).

We appreciate the reviewer's feedback and hope that these revisions better clarify the contributions and novelty of our study.

Fig-3 for reviewer

7. The data used for training/test in each analysis is not clear. For example, in the analysis related to Fig2, CGMega only uses one cell line for both training and test?

2.7 Response:

Sorry for the unclear description of this section. We used one cell line (MCF7) for performance evaluation and methods comparison (corresponding to Fig. 2a and Fig. 2b), and we also test the ability for few-shot transfer learning using two cell lines (MCF7 and K562) (corresponding to Fig. 2c).

For methods comparison, we performed CGMega and other methods on MCF7 cell line, which contain the most abundant omics data and gene labels. The training/test strategy is described in Q4 Response.

For few-shot transfer learning, we pre-trained CGMega on MCF7 cell line and fine-tune on K562 cell line with different numbers of label genes (Fig-1A for reviewer, related to Fig. 2c in the manuscript).

Beyond the above results, we guess that the reviewer may be confused by that the training and test set come from the same cell line. To address this possible concern, we trained CGMega on MCF7 cell line and test it on K562 cell line. As expected, the performance was relatively poor (CGMega test on K562 cell line, in Table-1 for reviewer). This is because that MCF7 cell line is derived from breast cancer donor and K562 cell is derived from leukaemia donor. These two cancers are different, resulting in different distributions of omics data. Thus, we recommend pre-train/fine-tune strategy for cross cell lines usage. As shown in Table-1 for reviewer and Fig. 2c in our manuscript, pre-trained CGMega is effective (AURPC increased from 0.7203 to 0.9155) even with a few labelled gene (AURPC = 0.8850, 100 labelled genes). This ability could satisfy most cancer research.

Table-1 for reviewer

	AUPRC
CGMega trained with MCF7, and test on K562 cell line	0.7203
Pre-trained CGMega with 1000 labeled genes from K562 cell line	0.9155
Pre-trained CGMega with 100 labelled genes from K562 cell line	0.8850

8. In the Data collection and preprocessing section, the author states "To validate the suitability of CGMega on other PPIs (Fig. S3a)..." (line 540). However, Fig. S3a doesn't seem to be about different other PPIs, which is a crucial comparison.

2.8 Response:

Thanks for pointing out this mistake, and we would like to clarify that the correct figure is Fig. S2f. We have corrected this in the revised manuscript.

9. The code repository needs to be polished for the work to be reproduced.

2.9 Response:

To facilitate reproducing the results of CGMega, we have further polished the tutorial notebook and uploaded both the original data and codes onto Zenodo (<https://zenodo.org/records/10086978>). Besides, a pre-built docker image and its Readme.md were also provided on Zenodo. The information includes file or directory that were previously missing now have been added. We wish our efforts may help reducing the workload for results replication.

Are you satisfied that all data and source code needed to reproduce the results of the paper have been made available?

The quality of the overall project looks good, but it still needs some revision. The tutorial notebook is great but it currently seems broken with missing datasets.

Are you satisfied that the results can be replicated using the code/software and dataset provided in the study?

I was not able to run the Tutorial notebook due to missing datasets.

Were you able to run the tool successfully?

No. First, some dependencies were missing (No module named 'torch_sparse'). I figured out by myself but then the tutorial notebook couldn't run because of missing datasets (No such file or directory: 'data/Breast Cancer Matrix/MCF7 Adjacent Matrix Ice').

Was the code sufficiently documented to allow another researcher to follow the algorithm?

There is some documentation, but they are not sufficient, especially for users who do not have experience of deep learning development. Also the current installation documentation on GitHub doesn't seem to support GPU. Additional cuda packages were required.

Can the software be run on a widely available operating system?

Although not tested, I think the software can be run on a wide range of operating systems, given the required dependencies are available across different operating systems.

To your knowledge, do available tools or software exist that perform in a similar way to the reported software?

Yes. I know the EMOGI work relatively well and I think EMOGI solves exactly the same questions as CGMega. CGMega has changed a few components in the model and shown superior predictive performance. But they are essentially very similar.

In cases when the source code is not provided but the mathematical description of the algorithm is; was the core mathematical algorithm sufficiently documented to allow another researcher to reproduce it?

Yes.

Reviewer #4 (Remarks to the Author):

The authors propose a method for predicting cancer genes by integrating multi-omics features. Besides, GNNExplainer is utilized for interpretability analysis, and some significant patterns are identified from gene modules in the context at either cell-line level or patient level. The prediction results demonstrate the effectiveness and robustness of the proposed model, and the explanations help us better understand the structure of cancer gene modules. This paper is interesting and well-written.

Response:

We appreciate the reviewer's efforts in helping us improving our work.

According to the issues from the reviewer, we improved our work from the following four aspects:

1. We have added more analysis to test the performance of our framework.
2. We have conducted inhibitor treatment experiments to investigate gene modules.
3. We have added detailed methods description and extended the discussion.
4. We have optimized and uploaded both the original data and codes onto Zenodo for further use.

All results have been added to the revised manuscript, and point-to-point responses are as below:

Major comments

*Q1. In page 12, the authors show a case study of detecting gene modules and important features of BRCA1 and BRCA2. While the goal here is to show that the predictions are interpretable because of some important patterns, but the **mechanistic explanations** are not supported clearly (e.g. only an intermediate gene of BRCA2 are related to cancer mechanism). It is suggested to further show how the functions of the detected genes are related to the cancer mechanisms, by providing more **in-depth explanations**.*

4.1 Response:

Thanks for pointing out our poor description of BRCA modules. To address this concern, we have made efforts in the following three areas:

(i) Deep analysis and discussion of important features.

(ii) Detailed explanations of BRCA2 gene modules with a dozen evidence.

(iii) Conducting tumor cells inhibition experiment with BRCA2 inhibitor and ROCK2 inhibitor treatment.

First, we examined known breast cancer genes with different important features. As shown in Fig-1 for reviewer, based on feature importance scores, genes from MCF7 cell line were generally divided into five clusters (by K-means clustering). Implications among these gene clusters are:

Many cancer genes (class-5) were as reported to be dominated by genetic mutations. Except for these, other genes were distinguished by the five Hi-C features, providing extended supplements based on their participations in each cluster: 1st, 4th and 5th Hi-C features showed joint effect with other regulatory factors on cancer driver genes (cluster-3), while 2nd and 3rd Hi-C features showed joint effect with genetic mutations (cluster-1). Some previous studies have verified our observations: (i) Gene *MYB* (in cluster-1) was reported to form fusion genes with *NFIB* due to the recurrent chromosomal translocation, which serves as a clear example of genotypic–phenotypic correlation for triple-negative breast cancer [1]. (ii) Dysregulation of gene *ADIPOR1* (in cluster-3) is widely observed in many cancers, but its genomic alteration frequencies are low [2]. This is consistent with CGMega that attributes HiC-1, HiC-5, chromatin accessibility and active histone modification H3K4me3 to *ADIPOR1*. (iii) Similar to *ADIPOR1*, gene *ALOX12* (in cluster-3) were significantly up-regulated in multiple breast cancer cell lines, which protect breast cancer cell from chemotherapy-induced growth arrest and apoptosis [3, 4], suggesting the importance of transcription regulation to *ALOX12*. (iv) Despite these isolated evidences, we collected RNA-seq data of breast cancers from TCGA project and identified differentially expressed genes (DEGs). The proportion of DEGs is the highest in cluster-3 (Fig-2 for reviewer). CGMega predicts HiC together with other active regulatory elements have joint effect of these genes. In addition, we also performed GO analysis on genes from cluster-2. positive regulation of gene expression and regulation of transcription from RNA polymerase II promoter were enriched in this gene set. **Together, feature importance scores provided by CGMega measure the joint effect of multiple factors, and can be used for guiding cancer genes classification.**

Fig-1 for reviewer.

Fig-2 for reviewer.

Second, more evidence implicate the potential mechanistic explanations for *BRCA2* gene module (*BRCA2* gene module is shown in Fig-3A for reviewer, and mechanistic explanations is shown in Fig-3B for reviewer). Among over one hundred of PPIs to *BRCA2*, the core part of *BRCA2* module is the interaction to *ROCK2*, which in turn connects with another 15 module genes. As reported, *ROCK2* is a candidate protein binding to *BRCA2* and complex consisting of *BRCA2* and *ROCK2* maintains the numerical integrity of centrosomes and accurate cell division [5]. Other genes in this module participate through nuclear export [6], signaling pathway such as *ROCK2/ADD1* signaling pathway [7], *Rho/ROCK* signaling pathway [8], and colocalization such as *EP300/ROCK2* colocalization [9]. Beyond these isolated cues, mechanism of other genes in *BRCA2* module is still unclear, also providing a candidate gene list or combination strategy for further experiments.

Fig-3 for reviewer

Finally, we focus on the relationship between *BRCA2* and *ROCK2*. We found that *ROCK2* expression was positively correlated with *BRCA2* expression in breast tumor donors while there is no such correlation in normal breast tissue (Fig-4 for reviewer, gene expression data were obtained from TCGA project). The co-expression of *BRCA2* and *ROCK2* in breast cancer suggest the joint effect in tumorigenesis, which may guide the effect enhancement of BRCA2 inhibitors on tumor cells.

To test this hypothesis, we conducted cellular biological experiment and treated MCF7 cells with BRCA2 inhibitor olaparib [10, 11] and with both BRCA2 inhibitor olaparib and ROCK2 inhibitor RKI-1447 [12]. Western Blot results have demonstrated the level of protein can be repressed after 24 hr, 48 hr, and 72 hr treatment with the combination of these two inhibitors (Fig-5A for reviewer). Then, we utilized Cell Counting Kit-8 (CCK-8) to determine the half maximal inhibitory concentration (IC₅₀) of olaparib (Fig-5B for reviewer) and olaparib/RKI-1447 combination (Fig-5C for reviewer). IC₅₀ value of inhibitors combination is lower than that of BRCA2 inhibitor alone, suggesting that combining ROCK2 inhibitor could enhance the sensitivity of BRCA2 inhibitor on MCF7 cells.

Together, by adding more analysis and cell drug experiment, we provided more clues for BRCA gene module and a trial for potential strategy by combining olaparib and RKI-1447 in breast cancer treatment.

Fig-4 for reviewer

Fig-5 for reviewer

References:

1. Martelotto, Luciano G et al., Genomic landscape of adenoid cystic carcinoma of the breast. *The Journal of pathology*, DOI:10.1002/path.4573
2. Zhuoyuan Chen et al., Distinct roles of ADIPOR1 and ADIPOR2: A pan-cancer analysis. *Front. Endocrinol.*, 2023, DOI: 10.3389/fendo.2023.1119534
3. Zhen Huang et al., ALOX12 inhibition sensitizes breast cancer to chemotherapy via AMPK

- activation and inhibition of lipid synthesis. *Biochem Biophys Res Commun.*, 2019, DOI: 10.1016/j.bbrc.2019.04.101
4. Siyuan Weng et al., ALOX12: A Novel Insight in Bevacizumab Response, Immunotherapy Effect, and Prognosis of Colorectal Cancer. *Front. Immunol.*, 2022, DOI: 10.3389/fimmu.2022.910582
 5. Huifeng Wang et al., BRCA2 and Nucleophosmin Coregulate Centrosome Amplification and Form a Complex with the Rho Effector Kinase ROCK2. *Cancer Research*, 2011, DOI: 10.1158/0008-5472.CAN-10-0030
 6. D A Freedman et al., Nuclear Export Is Required for Degradation of Endogenous p53 by MDM2 and Human Papillomavirus E6. *Molecular and Cellular Biology*, 1998, DOI: 10.1128/MCB.18.12.7288
 7. Kai Zheng et al., miR-135a-5p mediates memory and synaptic impairments via the Rock2/Adducin1 signaling pathway in a mouse model of Alzheimer's disease. *Nature Communications*, 2021, DOI: 10.1038/s41467-021-22196-y
 8. Imola Wilhelm et al., Role of Rho/ROCK signaling in the interaction of melanoma cells with the blood brain barrier. *Pigment Cell Melanoma Res*, 2014, DOI: 10.1111/pcmr.12169
 9. Toru Tanaka et al., Nuclear Rho Kinase, ROCK2, Targets p300 Acetyltransferase, *Journal of Biological Chemistry*, 2006, DOI: 10.1074/jbc.M510954200
 10. Maaïke A C Bruin et al., Pharmacokinetics and Pharmacodynamics of PARP Inhibitors in Oncology. *Clin Pharmacokinet*, 2022, DOI: 10.1007/s40262-022-01167-6
 11. Jan-Willem Henning et al., Clinical Considerations for the Integration of Adjuvant Olaparib into Practice for Early Breast Cancer: A Canadian Perspective. *Current Oncology*, 2023, DOI: 10.3390/curroncol30080556
 12. Ronil A Patel et al., RKI-1447 is a potent inhibitor of the Rho-associated ROCK kinases with anti-invasive and anti-tumor activities in breast cancer. *Cancer Research*, 2012, DOI: 10.1158/0008-5472.CAN-12-0954

Q2. In page 18, the authors identified some significant high-order patterns of AML gene modules, but it's unclear how these patterns are related with the AML. Therefore, it is suggested to conduct more analyses about mechanisms of AML disease, in order to mine more insightful patterns and further prove the interpretability of the method.

4.2 Response:

Thanks for this advice. To address this concern, we have conducted more analysis on the high-order of AML gene modules.

Since complex diseases, such as cancer, are both polygenic and multifactorial, dismantling the higher order structure of gene modules is necessary for network-based analysis ^[1]. Hubs of high-order gene modules have strong ties to their neighborhood, getting damaged could break the system into small, nonfunctional elements ^[2]. Identifying hub genes in gene modules has led to the identification of several gene essential in cancer ^[3,4], type 2 diabetes ^[5], chronic fatigue ^[6], other diseases ^[7,8] and tissue regeneration ^[9].

In our work, we described two high-order patterns (In page 18) including hub locations of gene module (Fig. S5b) and two-hop gene module (Fig. 6e). We first extended analysis corresponding to Fig. S5b, which depicts the hub location of *ESR1* in many known AML gene modules. As shown in Fig-6 for reviewer, totally 12 known driver genes and 5 predicted AML genes were identified as hub genes, which participates in ≥ 20 cancer gene modules. *EGFR*, *MYC*, *TP53*, *MAPK1*, and *PIK3R1* were well-known genes in cancer pathway^[10-14]. *EP300*, *CREBBP*, and *STAT3* were used as clinical testing gene panel for myeloid tumors^[15-17]. The successful detection of these genes as hub genes in all AML samples demonstrates the reliability of CGMega interpretation, and suggests the potential usage as AML gene panels of those five new hub genes including *ESR1*, *HDAC1*, *FYN*, *LYN*, and *GRB2*.

Fig-6 for reviewer

Next, we focus on the two-hop gene module (Fig-7A for reviewer), which is a supplementary pattern to cancer gene-centered gene module (Fig-7B for reviewer). For example, *BRCA1* gene module is cancer gene-centered pattern while *BRCA2* gene module and *KLF4* gene module in Patient 168 were two-hop pattern (See Fig. 4d and 6e in the revised manuscript). In two-hop gene modules, there often exists one key neighbor connecting the cancer gene and other genes. Such key neighbors (such as *ROCK2* in *BRCA2* gene module) help understand tumorigenesis and provide potential drug combination strategy (as discussed in Q1 response). We found that the two-hop pattern was widespread in AML samples, covering about 1/3 of all AML driver gene modules (Fig-7C for reviewer). Among these two-hop gene modules, the key neighbors (that connects over a half of genes in module) consists of known cancer genes, predicted cancer gene and other genes (Fig-7D for reviewer). Inspired by the *BRCA2*-*ROCK2* pair, we next focus on cancer gene and key neighbor pairs in AML samples. Total 142 pairs were conserved in over four samples, and several pairs were

highly conserved in all AML samples (Fig-7E for reviewer). We then performed GO analysis using both the cancer genes and key neighbor genes in these 142 pairs, and found that, different from cancer genes, the key neighbor genes were significantly enriched in signaling processes such as signal transduction and signaling pathway (Fig-7F for reviewer), suggesting that genes participating in signal processes may be the regulator or collaborator of known cancer genes.

Fig-7 for reviewer

Thanks for the helpful advice in reminding us to deeply deal with the high-order pattern of gene modules, and we have added the analysis and discussion in the revised manuscript.

References:

1. Christopher El Hadi et al., Polygenic and Network-based studies in risk identification and demystification of cancer. *Expert Rev Mol Diagn*, 2022, DOI: 10.1080/14737159.2022.2065195
2. Albert R et al., Error and attack tolerance of complex networks. *Nature*, 2000, DOI: 10.1038/35019019

3. Chou WC et al., Visual gene-network analysis reveals the cancer gene co-expression in human endometrial cancer. *BMC Genomics*, 2014, DOI: 10.1186/1471-2164-15-300
4. Oh EY et al., Extensive rewiring of epithelial-stromal co-expression networks in breast cancer. *Genome Biology*, 2015, DOI: 10.1186/s13059-015-0675-4
5. Keller MP et al., A gene expression network model of type 2 diabetes links cell cycle regulation in islets with diabetes susceptibility. *Genome Research*, 2008, DOI: 10.1101/gr.074914.107
6. Presson AP et al., Integrated weighted gene co-expression network analysis with an application to chronic fatigue syndrome. *BMC Syst Biol*, 2009, DOI: 10.1186/1752-0509-2-95
7. Voineagu I et al., Transcriptomic analysis of autistic brain reveals convergent molecular pathology, *Nature*, 2011, DOI: 10.1038/nature10110
8. Zhao W et al., Weighted gene coexpression network analysis: state of the art. *J Biopharm Stat*, 2010, DOI: 10.1080/10543400903572753
9. Rodius S et al., Analysis of the dynamic co-expression network of heart regeneration in the zebrafish. *Scientific Reports*, 2016, DOI: 10.1038/srep26822
10. Chong, Curtis R, and Pasi A Jänne. The quest to overcome resistance to EGFR-targeted therapies in cancer. *Nature medicine* 2013, DOI:10.1038/nm.3388
11. Dhanasekaran, Renumathy et al. The MYC oncogene - the grand orchestrator of cancer growth and immune evasion. *Nature reviews. Clinical oncology*, 2022, DOI:10.1038/s41571-021-00549-2
12. Giacomelli, Andrew O et al. Mutational processes shape the landscape of TP53 mutations in human cancer. *Nature genetics*, 2018, DOI:10.1038/s41588-018-0204-y
13. Tianlu Jiang et al. A novel protein encoded by circMAPK1 inhibits progression of gastric cancer by suppressing activation of MAPK signaling. *Molecular cancer*, 2021, DOI:10.1186/s12943-021-01358-y
14. Xiaoxu Huang et al. Circular RNA AKT3 upregulates PIK3R1 to enhance cisplatin resistance in gastric cancer via miR-198 suppression. *Molecular cancer*, 2019, DOI:10.1186/s12943-019-0969-3
15. Warren Fiskus et al. Targeting of epigenetic co-dependencies enhances anti-AML efficacy of Menin inhibitor in AML with MLL1-r or mutant NPM1. *Blood cancer journal*, DOI: 10.1038/s41408-023-00826-6
16. Yue Zhu et al. Oncogenic Mutations and Tumor Microenvironment Alterations of Older Patients With Diffuse Large B-Cell Lymphoma. *Frontiers in immunology*, DOI: 10.3389/fimmu.2022.842439
17. Chaoxiong Wang et al. CD300ld on neutrophils is required for tumour-driven immune suppression. *Nature*, DOI: 10.1038/s41586-023-06511-9

Q3. The authors conduct interpretability analysis using GNNExplainer. Since the graph transformer layers with attention mechanism are used within the CGMega, I wonder why the authors do not use the learned attention scores of graph transformer layers for explanation. Would the analysis result be different from the explanations given by GNNExplainer?

4.3 Response:

Thanks for the expertise of your comment. We did not rely on attention mechanism to generate

explanation mainly due to the two reasons as following.

First, the issue of whether attention mechanisms provide interpretability is still controversial, and there is simply no agreement to be recognized. Taking the field of NLP as an example, Jain, S. *et al.* proposed that *Attention is not Explanation* ^[1]. Their experiments showed that different attention coefficients would lead to the same prediction, which provides a counterfactual argument for attention being not interpretable. However, another diametrically opposed view that *Attention is not not Explanation* was proposed by Wiegrefe, S. *et al* ^[2]. Although they provided almost a point-to-point rebuttal to the former work, they did not prove that attention is interpretable (none of the other published research has proven or disproven it either). **Besides, they believed that the attention score used to provide interpretability are not unique, which challenges the well-recognized rule that the underlying mechanisms of a cancer gene in a specific cancer type are usually clear and definite rather than multiple or ambiguous.**

Second, the attention scores are learned to optimize the overall performance, and each score quantifies to what degree the attention should be paid by the model to the corresponding edge (i.e., a specific protein-protein interaction). However, all the scores are learned through the whole training process and therefore there were only one set of scores, which means that for any protein-protein interaction, attention mechanism only provides one fixed score. This is almost the opposite of the fact that one protein-protein interaction should play different roles with varied importance in the carcinogenic process driven by different cancer genes. In contrast with the model-level attention scores, **GNNExplainer generates its explanation for gene-module by masking edge one at a time, thus the importance scores for one edge in different gene-modules are accordingly various.**

As the reviewer suggested, **we calculated the attention scores in our framework and compared them with GNNExplainer's importance scores on MCF7 cell line. Results showed that using the above two ways will lead to interpretations with far more differences than similarities.**

1. From the perspective of edge, Table-1 for reviewer shows the example of edge *SMAD3-RUNX1*. According to the result of GNNExplainer, this edge has the highest importance score in the module of *RUNX1* while with trivial importance in the modules of several other genes. But attention mechanism only provides a fixed score for this edge and does not support any gene-specific analysis.

Table-1 for reviewer

Edge: SMAD3-RUNX1	GNNExplainer & score		Attention mechanism & score	
Module 1	RUNX1	0.7948	—	0.0037
Module 2	BRCA1	0.0423	—	0.0037
Module 3	BRCA2	0.0422	—	0.0037

Module 4	GATA3	0.0418	—	0.0037
Module 5	EGFR	0.0425	—	0.0037

2. From the perspective of gene, for example, as shown in Table-2 for reviewer, the top-1 edge in the module of *STAT3* calculated by GNNExplainer is *PTPRT-STAT3*. Among the attention scores for edges joining *STAT3* with other genes, the highest score is also for *PTPRT-STAT3*. Both GNNExplainer and attention mechanism all assign a rather important role to *PTPRT* (a phosphatase of the crucial *JAK-STAT* signaling pathway in anti-cancer immunity regulation ^[3]), indicating that something in common are learnt by the both. Whereas, the other edges at the top of their separate lists were all different and the distribution of two kinds of scores also varied a lot.

Table-2 for reviewer

Gene: STAT3	GNNExplainer & score		Attention mechanism & score	
Top-1 edge	PTPRT-STAT3	0.9615	PTPRT-STAT3	0.4172
Top-2 edge	PTPRT-EGFR	0.5097	SULT2A1-STAT3	0.3855
Top-3 edge	PTPRT-ATP2A2	0.0859	CHI3L1-STAT3	0.3232
Top-4 edge	STAT3-ROCK2	0.0520	HESX1-STAT3	0.3196
Top-5 edge	CSF3R-CSF3	0.0429	FRK-STAT3	0.2413
Top-6 edge	STAT6-STAT3	0.0429	TSLP-STAT3	0.2336
Top-7 edge	NFKB2-SIN3A	0.0428	IL23R-STAT3	0.2271
Top-8 edge	PTPRD-MTNRIA	0.0428	LRRFIP2-STAT3	0.1935
Top-9 edge	JAK2-CSF3R	0.0428	AZU1-STAT3	0.2025
Top-10 edge	PTPRT-ATP2A2	0.0859	SOCS7-STAT3	0.1800

Based on above reasons and analysis, we believe that using attention scores as explanations is not an appropriate choice while GNNExplainer owns its intrinsic strength more suitable for interpretation task on cancer gene.

References:

1. Jain, Sarthak and Byron C. Wallace. "Attention is not Explanation." *North American Chapter of the Association for Computational Linguistics* (2019)
2. Wiegrefe, Sarah and Yuval Pinter. "Attention is not not Explanation." *Conference on Empirical Methods in Natural Language Processing* (2019)
3. Shang, Xiaoling et al. "PTPRD/PTPRT mutation as a predictive biomarker of immune checkpoint inhibitors across multiple cancer types." *Front Immunol*, 2022, DOI: 10.3389/fimmu.2022.991091

Minor comments

Minor-Q1. From Figure 2e, SVM seems to perform better than MTGCN in terms of AUPRC, F1 score and ACC, so why do you say that MTGCN is the SOTA in page 6?

Minor-1 Response:

Thanks for reminding us. We meant to convey that we have compared with the most progressive methods in this field (i.e., MTGCN and EMOGI) at the time of writing. But we neglected that some basic machine learning methods as simple as SVM can also achieve good results and it was inappropriate to use ‘the state-of-the-art methods’ to describe MTGCN and EMOGI. In the revised manuscript, we have corrected this misstatement.

Minor-Q2. In Page 6, the sentence “we tested retrained CGMega (training from scratch) and pre-trained CGMega on K562 cell line using all labeled 164 genes” is not clear enough. I cannot figure out on which cell line the first model is retrained on and which cell line tested on, and whether the second model is fine-tuned or only is a pre-trained version. It is suggested to remove ‘tested’ and modify this sentence carefully.

Minor-2 Response:

Thanks for your kind suggestion. As the reviewer pointed, the vocabulary choice of ‘tested’ was misleading. We intended to describe that the first model was trained on K562 cell line (training from scratch), and the second model was fine-tuned on K562 cell line using its pre-trained version on MCF7 cell line. In the revised manuscript, we have modified the related sentences as following:

To this end, we adopted a two-step approach with CGMega. In the initial stage, CGMega was pretrained on the MCF7 cell line, allowing it to grasp fundamental patterns and characteristics prevalent in cancer genes. Following pre-training, we performed fine-tuning on other cancers, enabling CGMega to adapt and fine-tune its learned representations to the specific context of those rare cancers. To assess the performance of transfer learning, we conducted tests on the non-pretrained CGMega (trained from scratch) and the pretrained CGMega using all labeled genes (597 positives and 1839 negatives) in the K562 cell line.

Minor-Q3. In Figure S2b, it is interesting that the performance of the retrained CGMega does not exceed the pre-trained model. Can you explain more about this result?

Minor-3 Response:

Sorry for our misleading description in the previous manuscript. We believe this confusion derives from the same source as the above comment (minor-Q2). After the description being corrected, the message we were meant to show actually will be ‘the model trained from scratch does not exceed the fine-tuned version of the model’, and this is in line with our expectations.

Minor-Q4. In Figure S2e, you show the results of CGMega with different positive:negative ratios, but none of the results are matched with the original CGMega in Figure 2e. What is the positive:negative ratio you used for the original CGMega?

Minor-4 Response:

Thanks for reminding us to provide this detailed information in our manuscript. The collected dataset used for model training was highly imbalanced (positive:negative = 1:4 in MCF7 cell line). To prevent bias towards negative samples, we removed 50% of the negative samples, and finally trained CGMega on dataset with positive:negative ratio of 1:2.2. We have added this information in the revised manuscript.

In Fig. S2e. to examine the robustness of our model to different positive-to-negative sample ratios, we conducted comparative experiments with different ratios. Although the generated datasets were imbalanced, CGMega's performance remained relatively stable, indicating its robustness to variations in the positive-to-negative sample ratio.

Minor-Q5. Will CGMega still outperform the best baseline if you use different positive:negative training ratios (in Figure S2e) or different PPI databases (Figure S2f)?

Minor-5 Response:

To address this concern, we performed methods comparison on different P:N ratios and different PPI databases.

As shown in Fig-8 for reviewer, when considering various positive and negative sample ratios, CGMega maintains its position as the SOTA method. MTGCN and EMOGI, which have designed to alleviate unbalanced sample distributions, also exhibit robustness against varying sample ratios. GAT, on the other hand, leverages self-attention mechanisms to mitigate the impact of imbalanced samples by effectively balancing the contributions of neighboring nodes. However, GCN appears to be more susceptible to the variations in sample ratios, showcasing the importance of handling sample imbalance effectively.

Fig-8 for reviewer

Next, we test methods on five different PPI datasets as used in Fig. S2f. As shown in Table-3 for reviewer, CGMega performed best in four PPI datasets except Multinet dataset. MTGCN surpasses CGMega when evaluated on Multinet, which contains the fewest PPIs exist in this dataset, resulting in a sparsity of 0.03% (83766/16165²). This observation implies that CGMega excels at learning in dense network environments. MTGCN’s performance on Multinet may be due to its auxiliary task of PPI link prediction. MTGCN can utilize information from isolated nodes by incorporating a reconstruction loss.

Table-3 for reviewer

AURPC on different PPI datasets					
PPI dataset	CGMega	MTGCN	EMOGI	GAT	GCN
CPDB	0.9140	0.8290	0.7393	0.7752	0.6976
STRING	0.8953	0.8332	0.5304	0.8495	0.4955
iRef	0.8659	0.7451	0.6387	0.5919	0.5672
PCNet	0.8955	0.8462	0.8113	0.7340	0.7471
Multinet	0.8062	0.8545	0.6151	0.7037	0.5739

Reviewer #1 (Remarks to the Author):

The authors addressed the comments and the manuscript is well improved.

Minor comment:

- The authors described the process of deciding the 10-fold threshold of representative features in the rebuttal. It would be better if it is briefly mentioned in the manuscript as well.

Reviewer #1 (Remarks on code availability):

Docker image and tutorial is provided to run the code with same system configuration as the authors used, and the code is well organized. It requires a GPU with a large memory (more than 24GB) and the code itself is large (16GB) as well, which might be hard for individual users to run the code.

Reviewer #2 (Remarks to the Author):

The authors conducted a wide range of additional analyses and experiments. However, some of my concerns are still not adequately addressed.

In Response 2.4, the author said Figure 2 and S2 do not report cross-validation results. However, the descriptions in both the response and the revised manuscript suggest the implementation of cross-validation. It appears that the authors are referring to the validation fold as the "test set", which differs from the external test set that I recommended. Testing the model on an entirely separate dataset, in addition to partitioning a single dataset into training and test sets, is essential.

Response 2.7 addresses this to some extent by evaluating the model on a different cell line than the one utilised for training. Yet, the model demonstrates suboptimal performance without pretraining. The authors claimed this was due to the cell line being of another tissue type. It would then be important to test the model on another breast cancer cell line so that results are comparable.

I appreciate the additional comparison with MODIG as requested by the other reviewer. However, I don't think the message is clearly reflected in the main text. Fig2 only shows MODIG performs poorly, while the detailed discussion written in the response should also be mentioned.

Is Table-3 for reviewer based on overlapping features? In the original table in the MODIG paper, the performance is a lot better than 0.5 AUC even when a small subset of the features were used. I suspect some configurations were not properly set for MODIG so the performance was so poor.

Regarding the comparison with MODIG, my primary concern is the apparent novelty of CGMega, which, as noted by the authors in Response 2.6, seems primarily linked to the utilisation of different omics. Similar to CGMega's employment of Hi-C features, MODIG incorporates alternative structural features. The improvement in prediction accuracy with the addition of more features (or modalities) is an anticipated outcome. Therefore, I believe that the utilisation of a specific type of feature, such as Hi-C features, cannot be deemed the main contribution, unless the Hi-C data are primary data or necessitate a novel method for their incorporation into the model. The way CGMega uses Hi-C data also seems quite standard.

Other minor points:

Fig S2h seems to be coloured by each column, which could look confusing when comparing row-wise.

Reviewer #2 (Remarks on code availability):

NA

Reviewer #3 (Remarks to the Author):

I co-reviewed this manuscript with one of the reviewers who provided the listed reports as part of the Nature Communications initiative to facilitate training in peer review and appropriate recognition for co-reviewers.

Reviewer #4 (Remarks to the Author):

The authors have made extensive revisions with additional experiments, which are very impressive. But I still have one minor comment:

In Figure 5A for reviewer, is the result in the second row conducted with only ROCK2 inhibitor? If so, how to prove that the combination of both BRCA2 and ROCK2 inhibitors are effective using this figure?

Reviewer #4 (Remarks on code availability):

N/A

Reviewers' comments:

Reviewer #1 (Remarks to the Author):

The authors addressed the comments and the manuscript is well improved.

Response:

We thank the reviewer for his/her recognition of our effort in last revision.

Minor comment:

- The authors described the process of deciding the 10-fold threshold of representative features in the rebuttal. It would be better if it is briefly mentioned in the manuscript as well.

Response:

Thanks, and we have described this process in the Methods section of the revised manuscript.

Reviewer #2 (Remarks to the Author)

The authors conducted a wide range of additional analyses and experiments. However, some of my concerns are still not adequately addressed.

1. In Response 2.4, the author said Figure 2 and S2 do not report cross-validation results. However, the descriptions in both the response and the revised manuscript suggest the implementation of cross-validation. It appears that the authors are referring to the validation fold as the "test set", which differs from the external test set that I recommended. Testing the model on an entirely separate dataset, in addition to partitioning a single dataset into training and test sets, is essential.

2. Response 2.7 addresses this to some extent by evaluating the model on a different cell line than the one utilised for training. Yet, the model demonstrates suboptimal performance without pretraining. The authors claimed this was due to the cell line being of another tissue type. It would then be important to test the model on another breast cancer cell line so that results are comparable.

Response:

We understand the above two concerns about the model evaluation were from the construction and source of test sets. We incorporated these two questions to answer.

In our work, to conduct evaluation, we first constructed independent training set and test set as followings: 25% of the positive and negative genes were assigned to the test set while the remaining 75% were used for training. Then, we performed a standard procedure referred to as k-fold cross-validation on the training set and make prediction on the test set.

Here is a brief summary of this procedure:

1.Dataset Division: The training set is divided into 'k' equal parts or 'folds'. In our case, k is set to 10.

2.Model Training & Validation: For each individual fold, the model is trained on the remaining 'k-1' parts and validated on this leftover fold. This process is repeated 'k' times, such that each fold serves as the validation set once. This results in 'k' independently trained models.

3.Model Testing & Averaging: All 'k' models are then used to make predictions on the independent test set (the data that has never been exposed during the training process). The final prediction is the average of the predictions from all 'k' models, and was reported in Fig. 2 and Fig. S2.

Also, we acknowledge the importance about model performance on **another breast cancer cell line** as the reviewer pointed out. To test this, we collected additional Hi-C data and other omics data to construct new dataset for breast cell line, and ran CGMega on the new dataset. As shown in Table-1 for reviewer, the results of CGMega were comparable on different datasets. Details about these two datasets were shown in Table-2 for reviewer. This result helps demonstrate the stable performance of CGMega.

Thanks for helping us refine our work, and we have added these results in our revised manuscript.

Table-1 for reviewer

CGMega performance across different breast cancer datasets				
	AUPRC	AUROC	ACC	F1
Dataset-1 (Reported in the main text)	0.9140	0.9630	0.9216	0.8081
Dataset-2	0.9072	0.9627	0.9320	0.8272

Table-2 for reviewer

Data source of Dataset-1 and Dataset-2		
Dataset-1 is used in CGMega original paper and Dataset-2 is newly constructed for comparison		
	Dataset-1	Dataset-2
H3K4me3	ENCODE project ENCFF145CCI / ENCFF268RXB	ENCODE project ENCFF078BWS / ENCFF251QQR
H3K27ac	ENCODE project ENCFF340KSH / ENCFF491LQY	ENCODE project ENCFF054VCV / ENCFF071XTD
ATAC-seq	ENCODE project ENCFF821OEF	ENCODE project ENCFF976UNK
CTCF	ENCODE project ENCFF138LHE / ENCFF163JHE / ENCFF198YOJ	ENCODE project ENCFF157EYO / ENCFF237BZX / ENCFF915BMD
CNV	TCGA project 530 Breast cancer samples	TCGA project 106 randomly selected Breast cancer samples
SNV	TCGA project 4 Breast cancer samples	TCGA project 2 randomly selected Breast cancer samples
Hi-C	Genome Biology, 2015 GSE66733	Nucleic Acids Research, 2021 GSE182306

3. I appreciate the additional comparison with MODIG as requested by the other reviewer. However, I don't think the message is clearly reflected in the main text. Fig2 only shows MODIG performs poorly, while the detailed discussion written in the response should also be mentioned.

Response:

We appreciate the reviewer's recognition of our effort in comparison with MODIG. According to

the reviewer's helpful advice, we have added the detailed results and discussion in our newly revised manuscript.

4. Is Table-3 for reviewer based on overlapping features? In the original table in the MODIG paper, the performance is a lot better than 0.5 AUC even when a small subset of the features were used. I suspect some configurations were not properly set for MODIG so the performance was so poor.

Response:

We reported that the MODIG AUC achieved only 0.5001 with our dataset but in its original paper MODIG AUC was reported to achieve 0.8253. **We understand such a difference raises the reviewer's question that whether we used MODIG properly.**

This big difference of MODIG AUCs mainly derived from different datasets, especially due to the labelled genes in each dataset. In the last version of our response, we have only briefly described that CGMega is developed for cancer-specific genes while MODIG is for pan-cancer genes, but did not describe such difference between cancer-specific genes and pan-cancer genes in detail. We evaluated CGMega on breast cancer dataset, which was constructed based on the Network of Cancer Genes (NCG), Cancer Gene Census (CGC) along with high-confidence (level ≥ 0.95). **There were distinct differences between breast cancer-specific genes and pan-cancer genes as MODIG used (Table-1 for reviewer).**

Moreover, to further clarify that we have configured MODIG properly, we performed addition ablation experiment for MODIG (See Table-3 for reviewer). This ablation experiment was performed on the pan-cancer dataset that MODIG used which includes 2983 pan-cancer genes. In Test-1, Test-3, Test-4, and Test-5, 1939 labelled genes were randomly selected to make a fair comparison as on data of breast cancer cell line. As shown in Table-4 for reviewer, **we have reproduced MODIG performance** (See last two columns), and we also observed effects of node features on MODIG performance.

Thanks for reminding us to discuss the difference of MODIG performance on our dataset and MODIG dataset, and we have added this discussion in our newly revised manuscript.

Table-3 for reviewer

Comparison of MODIG dataset and CGMega dataset			
	Pan-cancer genes MODIG dataset	Breast cancer genes CGMega dataset	Overlap
All labels	2983	1939	612

Positive labels	796	358	188
Negative labels	2187	1581	417
P : N ratio	1 : 2.7	1 : 4.4	1 : 2.2

Table-4 for reviewer

Performance of MODIG on different datasets							
Input data	Test 1	Test 2	Test 3	Test 4	Test 5	Test 6	MODIG paper
Number of labelled genes	1939	2983	1939	1939	1939	2983	2983
Node features	Somatic Mutation	✓	✓	✓	✓	✓	✓
	CNV	✓	✓	✓	✓	✓	✓
	DNA methylation	-	-	✓	-	✓	✓
	Gene expression	-	-	-	✓	✓	✓
PPI	✓	✓	✓	✓	✓	✓	✓
MODIG AUROC	0.7603	0.7863	0.7812	0.7826	0.8001	0.8187	0.8243

5. Regarding the comparison with MODIG, my primary concern is the apparent novelty of CGMega, which, as noted by the authors in Response 2.6, seems primarily linked to the utilisation of different omics. Similar to CGMega's employment of Hi-C features, MODIG incorporates alternative structural features. The improvement in prediction accuracy with the addition of more features (or modalities) is an anticipated outcome. Therefore, I believe that the utilisation of a specific type of feature, such as Hi-C features, cannot be deemed the main contribution, unless the Hi-C data are primary data or necessitate a novel method for their incorporation into the model. The way CGMega uses Hi-C data also seems quite standard.

Response:

Actually as the reviewer pointed out, prediction accuracy will be improved when using additional biological features, this improvement was also observed in MODIG. However, Hi-C features is special compared to other features. First, Hi-C data have high dimension and low signal-to-noise ratio compared to other features such DNA sequence and ChIP-seq density. Second, Hi-C features are highly cell-type specific compared to other structural features such as PPI, GO, and Pathway. Third, chromatin structure information provides an independent view and is necessary to understand cancer mechanism (We have summarized this progress in a recent review, Junting Wang et al., *Quantitative Biology*, 2022).

Together, employment of Hi-C features is not simple and there is no standard procedure has been provided so far. For example, we found that directly constructing a graph using contacts from Hi-C

data is not suitable because the sparsity and noisy of Hi-C data will lead to lots of false-positive or isolated nodes, and will further destroy model performance. Thus, one of our main contributions in this work is to design and test different Hi-C features incorporations into the model. These detailed efforts have not been done before. Our strategy of combining Hi-C features with other features is not only successful in cancer genes prediction, but also provides a guideline for utilization of the ever-increasing Hi-C data and even single-cell Hi-C data.

Other minor points:

Fig S2h seems to be coloured by each column, which could look confusing when comparing row-wise.

Response:

Thanks for the careful review. We have modified the manner of colouring for performance comparison as follows.

	CGMega	MTGCN	EMOGI	GAT	GCN
CPDB -	0.9140	0.8290	0.7419	0.7752	0.6876
STRING -	0.8953	0.8332	0.5304	0.8495	0.4955
iRef -	0.8659	0.7451	0.6387	0.5919	0.5672
PCNet -	0.8955	0.8462	0.8113	0.7340	0.7471
Multinet -	0.8062	0.8545	0.6151	0.7037	0.5739

Reviewer #3 (Remarks to the Author)

I co-reviewed this manuscript with one of the reviewers who provided the listed reports as part of the Nature Communications initiative to facilitate training in peer review and appropriate recognition for co-reviewers.

Response:

We appreciate the time and energies that the reviewer poured into testing our framework.

Reviewer #4 (Remarks to the Author)

The authors have made extensive revisions with additional experiments, which are very impressive. But I still have one minor comment:

Response:

We thank the reviewer for his/her recognition of our effort in last revision.

1. In Figure 5A for reviewer, is the result in the second row conducted with only ROCK2 inhibitor? If so, how to prove that the combination of both BRCA2 and ROCK2 inhibitors are effective using this figure?

Response:

We are sorry for this unclear description in our last revision. The second row in Fig. 5A is the result with only ROCK inhibitor. We used this figure to show that ROCK2 protein level is repressed when treating with RKI-1447.

To test the effectiveness of combination of two inhibitors, we followed the general two-steps inhibitors combination experiments: **First**, we need to confirm that we have used the inhibitors in the right way. This can be done with Western Blot experiments to test whether the related protein have been repressed. **Then**, after this confirmation, we can test whether two inhibitors' combination is effective to inhibit tumor cell proliferation.

Accordingly, in our work, we first treated MCF7 cell line with BRCA2 inhibitor olaparib and ROCK2 inhibitor RKI-1447, respectively. Western Blot results show that olaparib is effective to repress BRCA2 level (first row in Fig. 5A for reviewer) and RKI-1447 is effective to repress ROCK2 level (second row in Fig. 5A for reviewer). After confirming this, we then test the effect of these two inhibitors combination by utilizing Cell Counting Kit-8 (CCK-8) to determine the IC50, and our results showed that combining ROCK2 inhibitor could enhance the sensitivity of BRCA2 inhibitor on MCF7 cells (Fig. 5C for reviewer).

Fig.5 for reviewer

Reviewer #2 (Remarks to the Author):

The authors have addressed all my concerns.

Reviewer #3 (Remarks to the Author):

My concerns have been addressed and I think the manuscript is ready to be published.

Reviewer #4 (Remarks to the Author):

The authors have explained the second row of Fig. 5A for reviewer clearly. However, when comparing Fig. 5B and Fig. 5C for reviewer, I can't be convinced about this claim made by the authors in the main text as well as the response letter: "IC50 value of inhibitors combination is lower than that of BRCA2 inhibitor alone, suggesting combining ROCK2 inhibitor could enhance the sensitivity of BRCA2 inhibitor on MCF7 cells."

First, the treatments with combination drugs lead to only slight lower IC50 than using Olaparib alone. I wonder if this difference can be significant and robust, not mention clinically effective.

Secondly, for the two left-most plots (24 hr), the combination treatment has slightly more significant lower IC50 and slope than using only Olaparib. However, the left-most figure of Fig. 5B (for reviewer) is a bit misleading. The highest value of vertical bar should be 80 like other plots, rather than 60 which would make the slope more steep. It is also suspicious that the starting IC50 value of the left-most plot of Fig. 5B is much lower than 20, which also makes the curve look more steep.

Overall, the signals in comparing the two rows of plots are not strong enough to support the main conclusion that the drug combination leads to lower IC50. The authors need to double check the experimental results, and give some clarification and discussion.

Reviewer #4 (Remarks on code availability):

N/A

Reviewers' comments:

Reviewer #4 (Remarks to the Author):

The authors have explained the second row of Fig. 5A for reviewer clearly. However, when comparing Fig. 5B and Fig. 5C for reviewer, I can't be convinced about this claim made by the authors in the main text as well as the response letter: "IC₅₀ value of inhibitors combination is lower than that of BRCA2 inhibitor alone, suggesting combining ROCK2 inhibitor could enhance the sensitivity of BRCA2 inhibitor on MCF7 cells."

First, the treatments with combination drugs lead to only slight lower IC₅₀ than using Olaparib alone. I wonder if this difference can be significant and robust, not mention clinically effective. Secondly, for the two left-most plots (24 hr), the combination treatment has slightly more significant lower IC₅₀ and slope than using only Olaparib. However, the left-most figure of Fig. 5B (for reviewer) is a bit misleading. The highest value of vertical bar should be 80 like other plots, rather than 60 which would make the slope more steep. It is also suspicious that the starting IC₅₀ value of the left-most plot of Fig. 5B is much lower than 20, which also makes the curve look more steep.

Overall, the signals in comparing the two rows of plots are not strong enough to support the main conclusion that the drug combination leads to lower IC₅₀. The authors need to double check the experimental results, and give some clarification and discussion.

Response:

Thanks for pointing out the misleading figure of drug combination treatment. To address these concerns, we have revised the figure, added statistical analysis, and made rigorous claims as follows:

First, we have revised the range of vertical bar to fairly show the inhibition of different treatments (Fig. 1A for reviewer). As described in Methods of our manuscript, we repeated drug (or drugs combination) treatment experiments three times under 8 inhibitor dose points. The IC₅₀ and HillSlope values were calculated using GraphPad Prism, a software using for statistical analysis and graphical visualization. This calculation was done on LOG concentration values, and thus the starting value was calculated with 1 nM/L concentration (which is 0 with LOG operation). In this experiment, IC₅₀ is the estimated concentration when the inhibition rate is 50%, and low IC₅₀ value represents high inhibitor sensitivity. Moreover, we examined the difference in inhibition rates between these two conditions (Fig. 1B for reviewer). After 24 hr olaparib/RKI-1447 combination treatment, IC₅₀ value decreased from 6.437 to 5.418 and the inhibition rates were overall high compared to treated with olaparib alone. But this difference was slight after 48 hr and 72 hr

treatments.

Fig. 1 for reviewer

Then, we performed statistical analysis using GraphPad Prism to test whether the difference of inhibition rates is significant. For each of the 8 inhibitor doses, the mean inhibition rate was calculated based on three repeats data, generating a paired-sample dataset containing both the inhibition rates of olaparib alone and olaparib/RKI-1447 combination under different doses. Then, a paired *t*-test was performed on this paired dataset. As shown in Fig. 2 for reviewer (a statistical representation of Fig. 1B for reviewer, and we used this plot in our revised manuscript), the inhibition rates of olaparib combined with RKI-1447 is significantly higher than that of olaparib alone after 24 hr treatment (p value = 0.0023). But it was comparable between two groups after 48 hr and 72 hr treatment. In addition, as the reviewer pointed out, the starting value of the upper left-most plot of Fig. 1A for reviewer is much lower than 20, which may affect the significance. Although this value was the real observation of the low-dose (1 nM/L, which is 0 with LOG operation) olaparib treatment experiments, we still test the significance of removing this dose point (that is, using data from another 7 inhibitor doses instead of 8 inhibitor doses), and the difference remains

significant (p value was 0.0030).

Fig. 2 for reviewer

Together, above results showed that olaparib/RKI-1447 combination was more effective than using olaparib alone in inhibiting MCF7 tumor cells after 24 hr treatment, suggesting a potential strategy for enhancing BRCA2 inhibitor sensitivity. However, more experiments with finer time intervals and inevitably laborious work are required to investigate what is the best effective time of olaparib/RKI-1447 combination and why this effect of olaparib/RKI-1447 combination disappears after 48 hr.

According to the helpful advice from the reviewer and our new results, we have revised our claims more rigorously:

Previous version: "IC₅₀ value of inhibitors combination is lower than that of BRCA2 inhibitor alone, suggesting combining ROCK2 inhibitor could enhance the sensitivity of BRCA2 inhibitor on MCF7 cells"

Current version: "IC₅₀ value of inhibitors combination was lower than that of BRCA2 inhibitor alone. Moreover, the inhibition rates of olaparib combined with RKI-1447 were significantly higher than those of olaparib alone after 24hr treatment (p value = 0.0023, paired t-test). But it was comparable between two groups after 48hr and 72hr treatment. These results showed that the combination of BRCA2 and ROCK2 inhibitors was more effective than using BRCA2 inhibitor alone in inhibiting MCF7 tumor cells after 24 hr treatment, suggesting a potential strategy for enhancing BRCA2 inhibitor sensitivity."

Also, we have added new discussion in our revised manuscript as "Combination of ROCK2 inhibitor RKI-1447 provides a potential strategy for enhancing BRCA2 inhibitor sensitivity, but more experiments are required to investigate what is the best effective time of olaparib/RKI-1447 combination and why this effect of olaparib/RKI-1447 combination disappears after 48 hr."

All the above analysis and discussion have been added into our revised manuscript, and we truly appreciate the efforts from the reviewer to help improve our work.

Reviewer #4 (Remarks to the Author):

The authors' response is neat. I have no more comment. Thanks.

Reviewer #4 (Remarks on code availability):

N/A